**Subject Category:**
Biology (whole organism)

ecology/plant science

bryophytes, climate change, fungi, invertebrates, mosses, open top chamber

**Author for correspondence:**
Hannah M. Prather
e-mail: pratherh@reed.edu

†Present Address: Biology Department, Reed College, 3203 SE Woodstock Blvd. Portland, OR 97202-8199, USA.

# Species-specific effects of passive warming in an Antarctic moss system

Hannah M. Prather[1,†], Angélica Casanova-Katny[2], Andrew F. Clements[1], Matthew W. Chmielewski[1], Mehmet A. Balkan[1], Erin E. Shortlidge[1], Todd N. Rosenstiel[1] and Sarah M. Eppley[1]

[1]Center for Life in Extreme Environments and Department of Biology, Portland State University, 1719 SW 10th Avenue, SRTC Room 246, Portland, OR 97201, USA
[2]Laboratorio de Ecofisiologia Vegetal, Facultad de Recursos Naturales, Universidad Católica de Temuco, Rudecindo Ortega 03694, Temuco, Chile

HMP, 0000-0002-5345-5143; AC-K, 0000-0003-3860-1445;
MWC, 0000-0002-4467-4977; EES, 0000-0001-8753-1178;
SME, 0000-0002-2094-9454

Polar systems are experiencing rapid climate change and the high sensitivity of these Arctic and Antarctic ecosystems make them especially vulnerable to accelerated ecological transformation. In Antarctica, warming results in a mosaic of ice-free terrestrial habitats dominated by a diverse assemblage of cryptogamic plants (i.e. mosses and lichens). Although these plants provide key habitat for a wide array of microorganisms and invertebrates, we have little understanding of the interaction between trophic levels in this terrestrial ecosystem and whether there are functional effects of plant species on higher trophic levels that may alter with warming. Here, we used open top chambers on Fildes Peninsula, King George Island, Antarctica, to examine the effects of passive warming and moss species on the abiotic environment and ultimately on higher trophic levels. For the dominant mosses, *Polytrichastrum alpinum* and *Sanionia georgicouncinata*, we found species-specific effects on the abiotic environment, including moss canopy temperature and soil moisture. In addition, we found distinct shifts in sexual expression in *P. alpinum* plants under warming compared to mosses without warming, and invertebrate communities in this moss species were strongly correlated with plant reproduction. Mosses under warming had substantially larger total invertebrate communities, and some invertebrate taxa were influenced differentially by moss species. However, warmed moss plants showed lower fungal biomass than control moss plants, and fungal biomass differed between moss species. Our results indicate that continued warming may

impact the reproductive output of Antarctic moss species, potentially altering terrestrial ecosystems dynamics from the bottom up. Understanding these effects requires clarifying the foundational, mechanistic role that individual plant species play in mediating complex interactions in Antarctica's terrestrial food webs.

## 1. Introduction

Arctic and Antarctic ecosystems have experienced a rapid climate change although the rate of change has altered over time and the degree of climate change has been widely variable [1,2]. Polar ecosystems are uniquely sensitive to climate changes [2–6], and thus the biological impacts are likely to be more pronounced in these regions [2,7]. The Western Antarctic Peninsula and the Scotia Arc region of the Southern Ocean have been among the fastest warming polar regions [2,8]. Records show a 0.2°C increase per decade from 1950s to 1990s in the Scotia Arc region and an even greater increase of 0.54°C per decade recorded at the Faraday/Vernadsky Station on the western side of the Antarctic Peninsula (maritime Antarctica) [2]. Temperature increases have been highest in the winter along the Western Antarctic Peninsula, while summer warming has been greatest along the eastern Antarctic Peninsula. More recently (since the late 1990s), annual mean temperatures have stabilized along the Western Antarctic Peninsula due to natural fluctuations in atmospheric circulations rather than long-term climate change trends [9]. While long-term temperature trends on the Western Antarctic Peninsula are supported by decades of records from research stations [2], long-term precipitation data are hard to obtain as *in situ* measurements are difficult. Carbon isotope data from peat cores on Signy Island in the Scotia Arc region suggest that the climate in the past 50 plus years has become both warmer and wetter [10], presumably due to the increase in precipitation, particularly in the form of rain, as well as from glacier melt in summer months. Biodiversity, including plant diversity, in Antarctica is strongly driven by patterns of water availability [11–13], and the increase in water availability with climate changes will probably alter patterns of diversity and expose new potential habitats to be colonized by terrestrial biota, particularly pioneer species such as lichens and bryophytes [5,8].

Antarctica is mostly ice covered; of its 14 million $km^2$ surface area, only about 0.34% remains seasonally ice and snow free [14]. Most of the ice-free areas are small and 'island-like', leading to a diversity of terrestrial ecosystems [15]. Despite rapid warming in some regions, Antarctica remains one of the harshest environments on Earth, with relatively few terrestrial organisms able to survive on its ice-free terrain [16]. Antarctic vegetation is dominated by a cryptogam flora, with numerous species of lichens (more than 200 species) and bryophytes (more than 100 species) [17,18], and only two angiosperms. Within the cryptogam vegetation, the microfauna comprises terrestrial invertebrate groups, including tardigrades, nematodes, springtails, mites and dipterans [18,19]. Given the severe constraints of climate and habitable land surface area in Antarctica, an amelioration of any of these factors may have disproportionally large impacts on Antarctic terrestrial ecosystems [20].

Bryophytes, in particular, mosses, are a dominant terrestrial plant in Antarctica, particularly on the Western Antarctic Peninsula where they are the predominant land cover. Mosses and their associated microfauna serve fundamentally important roles in ecosystem functioning worldwide, contributing to the production of above-ground biomass [21,22] and regulation of abiotic conditions such as soil temperature and moisture [23], and are important drivers of ecosystem biogeochemical cycles [24–26]. Mosses provide habitat for a diversity of microorganisms and invertebrates that form complex food webs that regulate organic matter decomposition, carbon sequestration and nutrient cycling, providing an essential link between above-ground and below-ground ecosystem components [22,27]. In polar ecosystems, accumulating bryophyte biomass (land cover) provides high thermal insulation, water-holding and cation exchange capacities. Therefore, bryophyte cover in polar ecosystems exerts a strong influence on soil temperature, water regimes and nutrient cycling and provides a significant carbon sink in terms of global warming [23,28–30].

Little is known about the response of Antarctic mosses and their associated communities of microorganisms and invertebrates to climate change [31]; however, we can assume that the abundance and productivity of mosses will be influenced by coupled changes in temperature and precipitation in Antarctica. Mosses are poikilohydric organisms that rely directly on the environment for water and lack mechanisms to prevent desiccation, making them particularly sensitive to changes in climate and precipitation regimes at both micro- and macro-levels [32]. Mosses are often lumped together as a single group in Antarctic studies, but species-level functional traits can vary greatly, therefore looking at species-specific responses to warming will be important moving forward [21,33–35]. Warming on the Western

Antarctic Peninsula is anticipated to encourage the growth and spreading of mosses in Antarctica, which will further influence terrestrial ecosystem succession [8,25,36] and hold implications for the microorganism and invertebrate communities they sustain [8]. Moss growth can be clonal or sexual, and only a few studies on Antarctic mosses have examined the effects of warming on sexual reproduction [35,37], which will be key to understand changing moss communities. Furthermore, despite the important ecological connection of invertebrates and moss fertilization [38,39], no studies to date have examined both reproductive expression and associated invertebrate communities of the dominant moss species.

Given the magnitude of moss cover in ice-free maritime Antarctica, it is crucial to study species-specific responses to warming and how they may link to terrestrialization processes in the Antarctic. Here, we tested the effects of increased temperature on Antarctic moss communities using open top chambers (OTCs) established for 8 years (2008–2016) to simulate warming and control plots (unwarmed) at Juan Carlos Point on Fildes Peninsula, located in maritime Antarctica on King George Island (KGI). In the experiment, we examined vegetation species cover and collected cores of the two dominant moss species, *Polytrichastrum alpinum* and *Sanionia georgicouncinata*, which comprise 65% of the terrestrial vegetation cover in the area. By using these cores, we tested the effects of warming on moss canopy morphology, moss reproductive effort and moss-associated invertebrate communities and fungal biomass. We predicted that passive warming would differentially impact Antarctic moss species, highlighting underlying moss species-specific differences and that these differences would further scale to affect the Antarctic terrestrial moss-associated food web.

# 2. Material and methods

## 2.1. Study site and species

Our study site was located in the South Shetland Island archipelago on Fildes Peninsula, KGI, one of the largest ice-free areas in the region [17]. KGI is known for its rich bryophyte flora, containing 61 bryophyte species, of which 40 have been recorded on Fildes Peninsula [17]. Our warming experiment was carried out at Juan Carlos Point (62°12′ S, 58°59′ W), a moist coastal lowland site with northern exposure towards Drake Passage and a rich moss-grass community, a community type commonly found on several islands along the South Shetland Archipelago [37,40]. This community is dominated by the grass *Deschampsia antarctica* (Desv.) and two moss species, *P. alpinum* (Hedw.) and *S. georgicouncinata* (Hedw.). These two moss species have bipolar distributions and are widespread and common species in this region of Antarctica [17]. *Polytrichastrum alpinum* is dioecious, with separate male and female plants, while *S. georgicouncinata* is hermaphroditic, and both species are rarely sexually reproductive in Antarctica, with only a single record of plants with sporophytes for *S. georgicouncinata* [17]. The previous research from this same warming experiment demonstrated that *P. alpinum* increased sporophyte production with warming [37]. Similarly, a warming experiment at a nearby site found that gametangia increased significantly in *P. alpinum* plants in warmed versus control plots [35].

## 2.2. Passive warming experiment

As a part of the long-term warming experiment, we established 10 OTCs and matched controls in 2008 at Juan Carlos Point [37] (electronic supplementary material). OTCs are designed to produce an increase in air temperature by preventing the loss of heat by convection processes and have been commonly used in both Arctic and Antarctic ecosystem studies [41,42]. The OTCs used here are similar to those used in other warming studies located in Antarctica [35,37,43]. The chambers are composed of 3 mm thick, transparent acrylic panels, assembled into hexagonally shapes that taper to an open top of 40 cm height and a basal footprint of 106.4 cm$^2$. The acrylic walls have small perforations to allow air exchange and avoid excessive warming. The OTCs were placed on areas identified by researchers as having approximately 80–90% plant cover (with moss cover approx. 50% and the remaining percentage being lichen), and OTCs remained in place year-round. Matched control sites were marked within 1 m of each OTC and were selected with similar plant composition and cover, but did not receive treatment [37].

Previously, we have shown that the OTC treatment in this experiment significantly increased mean maximum daily air temperature during the study period (2008–2010), from 7.3°C in control plots to 10.5°C in OTCs [37]. Temperature and humidity data were collected using HOBO Pro v2 loggers (Onset Computer Corporation, Bourne, MA). However, the passive warming treatment had no significant effect on mean daily temperature or mean minimum daily air temperature (analysed annually or including

season as a factor; mean site minimum daily air temperature at Juan Carlos Point, −6.4°C). The highest warming effect measured was during the summer season, with an increase of 0.61°C inside the OTCs compared to control plots. These values are similar to other reported values for passive warming experiments in maritime Antarctica, where Bokhorst *et al.* [44] measured an increase of 0.7°C in annual mean temperature inside OTCs when compared to control plots. The use of OTCs also produced changes in microclimate as mean daily relative humidity was significantly lower in the OTCs (80.7%) than in control plots (91.7%) [37].

## 2.3. Moss species effects on abiotic conditions

To determine the impact of moss species identity on the microenvironment temperature within the warming experiment, moss canopy temperatures were measured during the austral summer of 2015. Moss canopy temperatures were measured in contiguous patches of each dominant moss species, *P. alpinum* and *S. georgicouncinata*, using thermocouple data loggers inserted approximately 0.5 cm into the moss canopy (HOBO, UX120-014M, Onset Computer Corporation, Bourne, MA), in both an OTC and a control plot. Recordings were made every 10 s from 9 to 18 January 2015.

To assess how moss species influence soil temperature and moisture, we constructed separate arrays of *P. alpinum* and *S. georgicouncinata*. Arrays consisted of 8 × 13 cm blocks of each moss species (with each block being the depth of the moss to the rhizoid level) paired with a control of bare soil. Soil temperature (°C) and volumetric water content ($m^3 m^{-3}$) were measured once an hour below moss species (2 cm below soil surface, below the rhizoid level) and in bare ground control plots (2 cm below soil surface) during the austral summer season in 2016 using Decagon EM50 and 5TM soil moisture and temperature sensors (Meter Group, Inc., Pullman, WA).

## 2.4. Plant and cryptogam community analysis

Plant and cryptogam communities were assessed by sight for the per cent cover in the summer season 2015, for seven pairs of OTC and control plots, using a 0.5 m square frame placed in permanently marked blocks within plots. Two researchers conducted the surveys, and all points were agreed upon to eliminate single observer bias. Lichens were grouped to genus level, and crustose forms were categorized by observable morphology. One vascular grass species, *D. antarctica*, was recorded in the experimental plots.

## 2.5. Moss canopy morphology

We quantified eight moss canopy characteristics by extracting 2 cm diameter cores of *P. alpinum* and *S. georgicouncinata* from each of five matched OTC and control plots in 2016. Intact cores, which included the entire moss profile from rhizoid to gametophyte top, were weighed to the nearest milligram (dry weight (DW)), and electronic calipers were used to measure the height of the intact core on randomly selected locations. Tissue height measurements (to the nearest millimetre) were conducted in four categories: photosynthetic tissue height, senescent tissue height, soil/rhizoid height and total core height. In addition, the height of five randomly selected gametophytes from each core was measured to the nearest millimetre to estimate the canopy height. Mean canopy height was calculated from the gametophyte measurements, and mean core height was calculated from the intact core measurements. Canopy density was determined by counting the number of gametophytes per core; leaves were denuded from one randomly selected gametophyte from each core to determine leaf density.

## 2.6. Reproductive biology and expressed sex ratios in *Polytrichastrum alpinum*

*Polytrichastrum alpinum* cores (2 cm diameter) from five matched OTC and control plots were separated, and each gametophyte was counted, confirmed as *P. alpinum* and categorized by age as juvenile (very green with leaves still appressed), mature (leafy gametophyte) or senescent (very little to no green tissue, blackening), following the methods of Shortlidge *et al.* [35], which are modified from a standard bryophyte protocol for placing leaves in age classes using colour and morphology [45]. Ten randomly selected gametophytes from the mature age category were dissected and examined for the presence of sex expression under compound and dissecting microscopes (Leica Application Suite 3.5.0, Leica, Germany). Examination of plants under the microscope is the standard protocol for determining sex expression for plants in this family [46]. We classified sex expression based on the presence or absence of male sex organs (antheridia), female sex organs (archegonia), sporophytes or paraphyses (sterile reproductive structure). If gametophytes had no

sexual expression, they were classified as purely vegetative (no evidence of gametangia, sporophytes or paraphyses). Laboratory determinations of sexually expressing gametophytes per core were combined and averaged for OTC and control plots.

## 2.7. Bryophyte-associated invertebrate communities

We quantified the invertebrate community (presence, absence and abundance) from the dominant moss species (*P. alpinum* and *S. georgicouncinata*) by extracting replicate 2 cm diameter cores from paired OTC and control plots, yielding 23 cores.

Invertebrates were extracted using an innovative dried substrate extraction technique, which was tested and found to yield comparable extraction numbers as matched extractions performed with Tullgren extractors from the same site location. Moss cores were placed directly into paper coin envelopes and dried in an oven at 30°C for 72 h. Each core was then carefully removed, placed into a weigh boat and core mass quantified to the nearest milligram. Cores were homogenized carefully by hand and divided into two 50 ml Falcon tubes. Invertebrates were extracted by vigorously shaking the tubes using a blend of kerosene and 95% ethanol (1 : 3 ratio). The solution was allowed to settle, and transfer pipettes were used to extract invertebrates from the kerosene suspension into a Petri plate for identification under dissecting microscope. Extracted invertebrates were sorted taxonomically into the following groups: Oribatida, Collembola and Nematoda. Invertebrate abundance was expressed as number of individuals per sampled moss core.

## 2.8. Estimation of bryophyte fungal biomass

Dried 2 cm diameter cores of *P. alpinum* and *S. georgicouncinata* were separated from the soil and lyophilized. For each sample, 100–200 mg of lyophilized tissue was placed in a 2 ml locking microcentrifuge tube with a zirconium bead and ground to a fine powder using a bead beater at 2500 r.p.m. for 40 s. Ergosterol was extracted from the powdered tissue following the methods of Dahlman *et al*. [47], with the following changes. Samples were suspended in 1 ml MeOH, agitated in an orbital shaker for 1 h at 320 r.p.m. in darkness and subsequently allowed to precipitate overnight at 4°C. Extractions were then centrifuged for 1 h at 14 000 r.p.m. and 4°C. Following centrifugation, the supernatant from each extraction was transferred to a fresh 2 ml microcentrifuge tube and centrifuged for an additional 10 min at 140 000 r.p.m. and 4°C. The supernatants from this secondary centrifugation were filtered through 0.2 μm acrodiscs into amber autoanalyser vials and stored at 4°C until high-performance liquid chromatography (HPLC) analysis. Extractions were analysed following [48] on a 1200 Series HPLC (Agilent Technologies, Waldbronn, Germany).

## 2.9. Statistical analyses

We used two-way analysis of variance (ANOVA) to determine the effect of treatment (OTC versus control), moss species (*P. alpinum* versus *S. georgicouncinata*) and the interaction between these factors on canopy temperature. We used ANOVA to determine the effects of ground cover type (*P. alpinum* versus *S. georgicouncinata* versus bare ground) on soil temperature and soil moisture. We employed a Tukey's honest significant difference (HSD) to examine pairwise contrasts among ground cover types. We analysed and visualized these data in the R statistical platform [49] version 3.3.3. with the packages: lubridate [50], ggplot2 [51] and dplyr [52].

To reveal potential changes in vegetation community assemblages between OTC and control plots, a per cent cover matrix was ordinated using non-metric multidimensional scaling (NMDS) in R studio [49] version 0.99.473 using package vegan [53]. By using Bray–Curtis distance measures and square root data transformation, we chose a two-dimensional solution with the best stress (0.09) based on 20 runs of 1000 permutations. An analysis of similarity (ANOSIM) was used to statistically test for differences between warmed and unwarmed cryptogam species assemblages.

We used a generalized linear model (GLM) to determine the effect of treatment (OTC versus control), bryophyte species (*P. alpinum* and *S. georgicouncinata*) and the interaction between these factors on core DW (with a normal distribution and an identity link function) and photosynthetic tissue height (with a Poisson distribution and a log link function) for the cores collected for each species for plant morphological analysis (separate bryophyte cores were collected for invertebrate analysis). For these analyses, we included subsite (the location of each paired OTC and control plot), but it was not significant and was dropped from the final analyses. We used *t*-tests to determine the effect of treatment on *P. alpinum* canopy density (number of gametophytes per core), number of juvenile

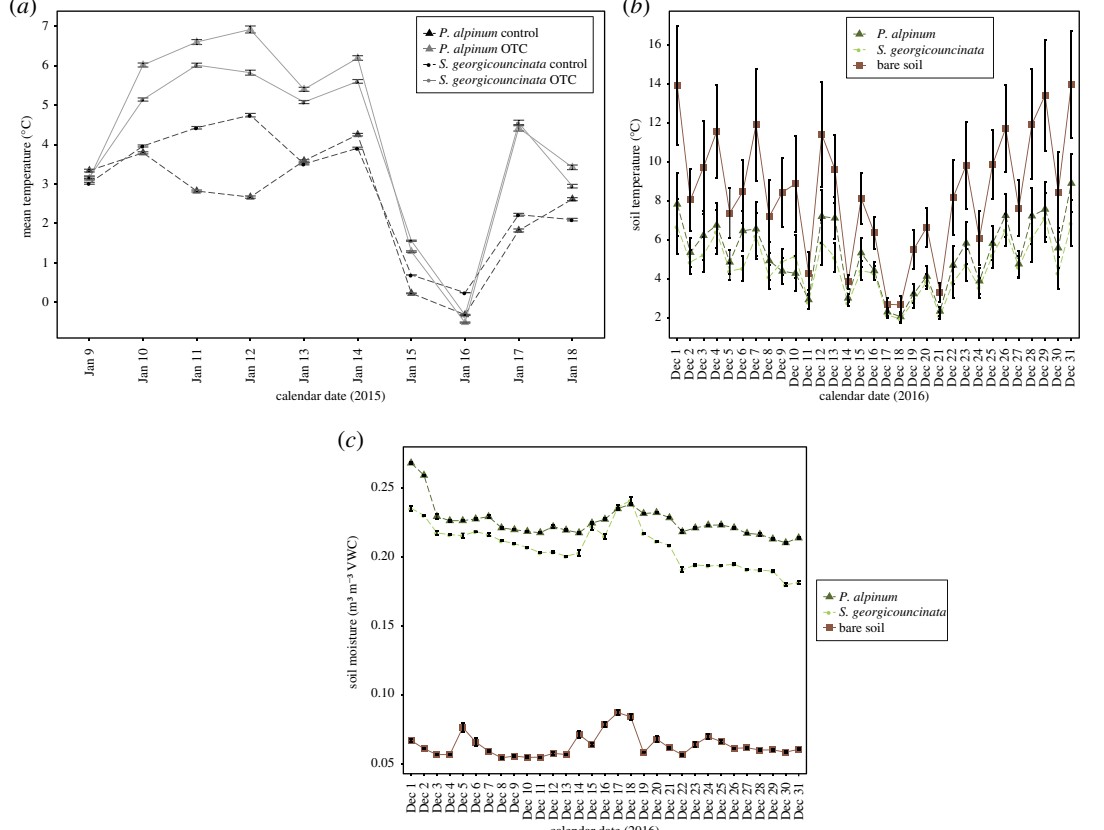

**Figure 1.** (*a*) Daily mean (±s.e.) moss canopy temperatures (°C) of control and OTC treatments of *P. alpinum* and *S. georgicouncinata* field plots during austral summer 2015 (*n* = 86 400). (*b*) Daily mean (±s.e.) soil temperatures of *P. alpinum*, *S. georgicouncinata* and bare soil patches during austral summer 2016 (*n* = 744). (*c*) Daily mean (±s.e.) soil moisture (m³ m⁻³ volumetric water content (VWC)) recorded below *P. alpinum*, *S. georgicouncinata* and bare soil patches during austral summer 2016 (*n* = 744).

gametophytes per core and gametophyte density (number of leaves per gametophyte), as these more intensive measures were only available for this species.

To determine the effect of treatment (OTC and control) and subsite (the location of each paired OTC and control plot) on gametophytes with female sex expression (archegonia) and gametophytes with sterile sex structures (paraphyses) in *P. alpinum*, we used a GLM with Poisson distribution with a log link to account for the large number of zeros in the dataset, which represent low-frequency effects rather than missing data values. No gametophytes with male sex expression were found.

We used simple regression analysis to determine whether nematode or microarthropod (Collembola and Oribatida) abundance was influenced by plant biomass. We used GLMs to test the effect of moss species (*P. alpinum* and *S. georgicouncinata*), treatment (OTC versus control), the interaction between these factors, subsite on the number of nematodes per bryophyte core (using a normal distribution and an identity link function) and the number of microarthropods per bryophyte core (using a Poisson distribution and log link function). We used a simple regression to determine whether microarthropod abundance was correlated with female sexual expression. We used a GLM to test the effect of moss species (*P. alpinum* and *S. georgicouncinata*), treatment (OTC versus control) and the interaction between these factors on ergosterol content, used as a proxy for bryophyte fungal biomass. All analyses were conducted using JMP version 14.1.0 [54], unless otherwise stated.

# 3. Results

## 3.1. Moss effects on abiotic conditions

Temperatures in both *P. alpinum* and *S. georgicouncinata* mosses were significantly higher in OTC compared to control plots (*n* = 172,800, *F* = 9906.59, *p* < 0.0001, figure 1*a*). While *S. georgicouncinata*

control plots were significantly warmer than *P. alpinum* controls ($n = 86\,400$, $F = 5.51$, $p = 0.02$, figure 1*a*), the opposite was true for OTC treatments. The interaction of treatment and species was also significant ($n = 86\,400$, $F = 643.73$, $p < 0.0001$, figure 1*a*).

Bare soils were significantly warmer than soils under either *P. alpinum* or *S. georgicouncinata*, in addition, moss identity did not significantly impact soil temperatures ($n = 744$, $F = 68.35$, $p < 0.0001$, figure 1*b*). Soil moisture content was drastically lower in bare soil treatments than in soil under either moss species ($n = 744$, $F = 32\,546.00$, $p < 0.0001$, figure 1*c*). *Polytrichastrum alpinum* cover provided a significant increase in soil moisture when compared with soils covered by *S. georgicouncinata* or in bare soil ($n = 744$, F = 32 546.00, $p < 0.0001$, figure 1*c*).

## 3.2. Cryptogam community response to passive warming

Across the seven OTC and control plots surveyed, we found seven bryophyte species, *Bartramia patens*, *Bryum pseudotriquetrum*, *Meesia uliginosa*, *Pohlia* sp., *P. alpinum*, *S. georgicouncinata* and *Syntrichia saxicola*. Two species of bryophytes occurred only in OTC plots, *B. patens* and *Pohlia* sp. In addition, NMDS ordination of cryptogam communities showed that community composition showed no detectable compositional changes (as measured by the per cent cover) after 8 years of passive warming ($n = 7$, NMDS stress = 0.09, ANOSIM test statistic $p = 0.35$). All plots were abundant with vegetation, and no bare ground or rock substrate was recorded, which is as expected given the site lies in a moist, coastal lowland that is rich with cryptogamic vegetation.

## 3.3. Bryophyte canopy morphology under passive warming

Despite 8 years of passive warming, moss canopy morphology differed little between plants in control and warmed plots for the dominant moss species *P. alpinum* and *S. georgicouncinata*. Core DW was not significantly affected by the treatment (d.f. = 1; $X^2 = 2.61$; $p = 0.11$) or the interaction between treatment and species (d.f. = 1; $X^2 = 0.50$; $p = 0.48$). However, bryophyte species did differ significantly in DW (d.f. = 1; $X^2 = 6.18$; $p = 0.01$), with *P. alpinum* having cores of greater DW (2.12 g ± 0.33s.e.) than *S. georgicouncinata* (1.17 g ± 0.22s.e.). Similarly, photosynthetic tissue height was also not affected by the warming treatment (d.f. = 1; $X^2 = 0.83$; $p = 0.36$) or the interaction between treatment and species (d.f. = 1; $X^2 = 0.49$; $p = 0.49$). Photosynthetic tissue height did differ significantly between the two bryophyte species (d.f. = 1; $X^2 = 26.94$; $p < 0.0001$), with *P. alpinum* having taller photosynthetic gametophytes (8.33 mm ± 1.32s.e.) than *S. georgicouncinata* (3.33 mm ± 0.50s.e.).

For *P. alpinum*, in which more intensive morphological measurement were made, canopy density (number of gametophytes per core) did not differ between treatments (d.f. = 8; $t = -0.58$; $p = 0.58$). However, OTC plots contained significantly fewer juvenile stems than did control plots (d.f. = 8; $t = -2.97$; $p = 0.02$; 10.6 juvenile stems per core ± 1.97s.e. and 25.8 juvenile stems per core ± 4.73s.e., respectively). Leaf density (number of leaves per gametophyte) did not differ significantly between OTC and control plots (d.f. = 18; $t = -1.01$; $p = 0.33$).

## 3.4. Sex expression in *Polytrichastrum alpinum*

In *P. alpinum*, plants in OTCs produced female reproductive structures (archegonia) in significantly greater numbers than did plants in the control plots (d.f. = 1, $X^2 = 5.65$, $p = 0.02$, figure 2). Subsite (the location of the paired OTC and control plots) was significant in this analysis (d.f. = 4, $X^2 = 14.56$, $p = 0.006$). The effect of the warming treatment on the occurrence of paraphyses (sterile reproductive structures) was not significant (d.f. = 1, $X^2 = 3.80$, $p = 0.051$). No male sex expression (antheridia) or sporophytes were observed during the sampling season.

## 3.5. Bryophyte and invertebrate communities under passive warming

In extracted moss cores, nematodes were most abundant (23.17 individuals per core ± 3.74s.e.), followed by Collembola (2.39 individuals per core ± 0.50s.e.) and Oribatida (0.87 individuals per core ± 0.36s.e.). Invertebrate communities were affected by moss species, with significantly higher nematode abundance in *P. alpinum* cores than *S. georgicouncinata* cores ($n = 23$, d.f. = 1; $X^2 = 14.06$, $p < 0.0001$, figure 3*a*). Warming treatment (d.f. = 1; $X^2 = 0.00$, $p = 0.96$), the interaction of moss species and treatment (d.f. = 1; $X^2 = 1.17$, $p = 0.79$) and subsite (d.f. = 7; $X^2 = 7.89$, $p = 0.34$) had no significant effect on nematode abundance.

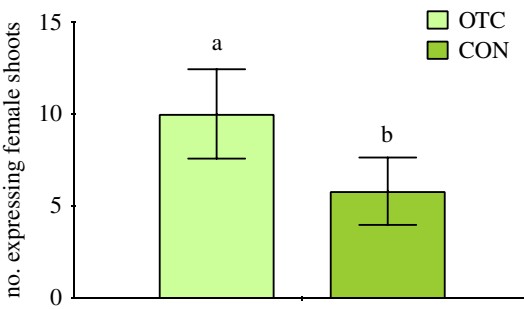

**Figure 2.** Mean (±s.e.) number expressing female reproductive structures in *P. alpinum* differed between warmed (OTC) and control (CON) plots. Warmed (OTC) plots exhibited more female sex expression than control plots (*n* = 50).

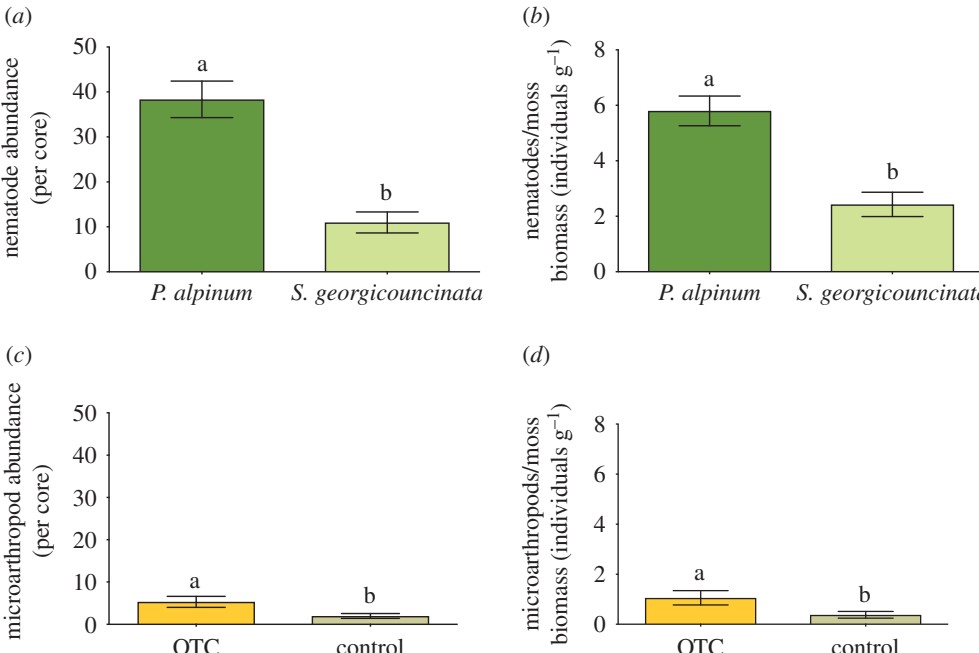

**Figure 3.** (*a*) Mean (±s.e.) nematode abundance from moss cores extracted from two dominant moss species *P. alpinum* and *S. georgicouncinata* expressed per bryophyte core (*n* = 23). (*b*) Mean (±s.e.) microarthropods abundance from moss cores extracted from two dominant moss species (*P. alpinum* and *S. georgicouncinata*) in warmed (OTC) and control (CON) plots expressed per bryophyte core (*n* = 23). (*c*) Mean (±s.e.) microarthropod abundance from moss cores extracted from two dominant moss species (*P. alpinum* and *S. georgicouncinata*) in warmed (OTC) and control (CON) plots expressed per bryophyte core (n = 23). (*d*) Mean (±s.e.) microarthropod abundance from moss cores extracted from two dominant moss species (*P. alpinum* and *S. georgicouncinata*) in warmed (OTC) and control (CON) plots expressed per moss biomass (individuals g$^{-1}$).

Invertebrate extractions from the dominant moss species, *P. alpinum* and *S. georgicouncinata*, revealed that OTC plots had significantly higher numbers of microarthropods (Oribatida and Collembola) than control plots (*n* = 23; d.f. = 1; $X^2$ = 12.45, *p* = 0.0004; figure 3*b*). Subsite was also a significant factor affecting microarthropod abundance (d.f. = 7; $X^2$ = 46.64, *p* < 0.0001). Moss species (d.f. = 1; $X^2$ = 0.26, *p* = 0.61) and the interaction of moss species and treatment (d.f. = 1; $X^2$ = 1.06, *p* = 0.30) had no significant effect on microarthropod abundance. In addition, we found that the total microarthropods (Oribatida and Collembola) were strongly and positively correlated with the number of female sex structures measured in *P. alpinum* (*n* = 10; $r^2$ = 0.67; *p* = 0.004).

## 3.6. Bryophyte fungal biomass under passive warming

We used ergosterol content as a proxy for fungal biomass in two dominant moss species and found that moss species significantly affected the bryophyte fungal biomass. Extracted cores of *P. alpinum* had significantly more fungal biomass than *S. georgicouncinata* (*n* = 12, d.f. = 1; $X^2$ = 20.00; *p* < 0.0001,

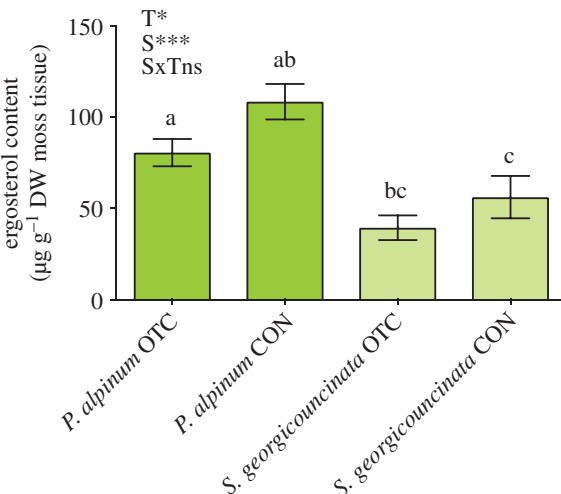

**Figure 4.** Mean (±s.e.) ergosterol content (μg g$^{-1}$ DW moss tissue) extracted from two dominant moss species (*P. alpinum* and *S. georgicouncinata*) in warmed (OTC) and control (CON) plots. Ergosterol content was significantly different among treatments (T) and *P. alpinum* had significantly more fungal biomass than *S. georgicouncinata* (S). The interaction between treatment and species (SxT) was not significant ($n = 12$).

figure 4), and contrary to our initial hypotheses, control plots had significantly higher fungal biomass than OTC plots (d.f. = 1; $X^2 = 6.42$, $p = 0.01$). The interaction between treatment and species was not significant (d.f. = 1; $X^2 = 0.46$, $p = 0.56$).

# 4. Discussion

Here, we present results of 8 years of experimental warming on a moss ecosystem in maritime Antarctica. Experimental warming had distinct and significant impacts on both microclimate conditions and bryophyte communities. We found that our OTC warming treatments were successful at significantly increasing air temperature during the study period, as they are designed to do, and in addition, moss canopy temperatures also significantly increased in OTC treatments. Relative humidity, as measured in air, significantly decreased in OTCs. Our abiotic results are similar to those generated in other Antarctic studies utilizing OTC chambers; see the study by Bokhorst *et al.* [42] for review on passive warming methods in the Antarctic. Yet, a large variation in abiotic conditions has been generated using OTCs in Antarctica, some of which are dissimilar to our findings. This variation demonstrates the variance in terrestrial, ice-free sites found in Antarctica and the varied methods of measurement (air, soil and moss canopy). Microsite differences are expected to be critical in explaining cryptogam survival during climate change in Antarctica, and more detailed understanding of topography and plant health is needed to understand this relationship [55–57].

Results from this passive warming experiment demonstrate that moss species were affected differently by warming and in turn differentially affected communities at the higher trophic level. Our results indicate that increasing warming in the Western Antarctic Peninsula will have distinct impacts on Antarctic moss sexual expression, and species-specific effects of these dominant land plants may hold implications for the terrestrial community of microorganisms and invertebrates that these mosses sustain. We discuss these results below.

## 4.1. Species-specific thermal effects observed in bryophytes under experimental warming

We found that mosses in OTCs had significantly higher canopy temperatures than those located in control plots, and significant species-specific canopy temperature differences also occurred, with higher canopy temperatures occurring in *P. alpinum* (figure 1*a*). These species-specific differences in canopy temperatures are probably due to albedo effects, as has been found in subarctic cryptogams [30]. Studies in higher latitudes suggest that moss species differ not only in their ability to absorb heat but also in their ability to transfer thermal energy to the soil [29]. We found that bare soils were significantly

warmer than when moss cover was present (figure 1*b*), but that moss species were similar in their insulating effects. Moss species did significantly affect soil moisture; *P. alpinum* provided a significant increase in soil moisture when compared with soils covered by *S. georgicouncinata* (figure 1*c*). Our results indicate that climate warming along the Western Antarctic Peninsula may alter the way that moss canopies influence energy exchange between the atmosphere and soils, with implications for the water–soil–atmosphere interface in a warming Antarctica. Moss cover can reduce the exchange of heat between the atmosphere and the soil, providing an insulating effect that decreases soil temperatures and protects permafrost [58,59] and further can provide habitat for microfauna adapted to the Antarctic ecosystem [60–62]. A reduction of surface cover of cryptogams or changes in the insulating and/or water-holding capacity of moss species may hold significant effects for future soil temperatures, soil moisture, thawing of permafrost, communities of microfauna and processes of ecosystem succession along the Western Antarctic Peninsula [23,25,58,63].

## 4.2. Effects of experimental warming on moss species

After 8 years of passive warming, we observed no distinguishable effects on cryptogam community composition overall, a finding consistent with other Antarctic passive warming studies [35,37,43,64]. However, we did find differences in sexual expression; *P. alpinum* in warmed (OTC) plots exhibited significantly more female sex expression than control plots. Our results are similar to those of Shortlidge *et al.* [35], who found increases in individual gametangia production and decreases in cellular stress defenses, suggesting that warming may relieve environmental constraints on reproductive expression. Our findings also identified a correlation between the number of female moss reproductive structures and the abundance of microarthropods within the moss canopy ($r^2 = 0.64$). Studies show that both springtails and mites prefer sexually expressing moss shoots rather than non-expressing shoots and that microarthropods prefer female over male shoots [38,39]. Microarthropods may prefer sexually expressing plants directly or indirectly (via fungi) because of sugars, starches and fatty acids excreted by gametangia [39,65,66]. Our study cannot determine whether microarthropods are increased on plants with gametangia because of these reproductive structures or because of an unmeasured factor that also correlates with sexual expression in our study. However, studies show that sexual reproduction in mosses is facilitated by springtails and mites [38,39], and because microarthropods may benefit more nutritionally from sexually expressing versus non-expressing plants [39,65,66], understanding the potential relationships among climate warming, sex expression in mosses and microarthropod and moss population dynamics will be instructive to understand the interaction between mosses and microarthropods in Antarctica.

## 4.3. Warming and moss species differentially affect higher trophic levels

Invertebrate communities extracted from bryophytes responded differentially under 8 years of passive warming treatment (OTCs) compared with controls. We found that overall microarthropod abundance increased in warmed (OTC) plots compared with control plots. These results are similar to those found by Day *et al.* [67] who found an increase in the abundance of the springtail *Cryptopygus antarcticus* under summer warming and water additions in the Antarctic Peninsula. However, mixed results have been common from warming experiments examining microarthropod abundance [68]. Bokhorst *et al.* [43] found little effect after 10 years of passive warming (OTCs) on invertebrate communities on Signy Island, while Convey *et al.* [69] showed significant declines in Collembola with warming as a result of desiccation. Ultimately, temperature-moisture regimes and microsite variation may have the most distinct influences on microarthropod communities in maritime Antarctica [67,69], and terrestrial site variation may be an important factor as invertebrates in wet sites may be differentially affected by warming experiments than those on drier or rocky sites in general.

While positive effects of warming were observed on microarthropod communities, moss species identity had a significant effect on abundance of soil nematodes. Our results show *P. alpinum* hosted a higher abundance of nematodes than *S. georgicouncinata*, demonstrating strong species-specific effects exist for the capacity of Antarctic mosses to host invertebrates. Mouratov *et al.* [70] found that the abundance of Antarctic soil nematodes on KGI was negatively correlated with soil water content. In addition, Mouratov *et al.* [70] found nematode abundance varied with plant host species, showing higher abundances under *D. antarctica* when compared with *S. georgicouncinata*, similar to findings in our study that revealed higher abundances under *P. alpinum* than *S. georgicouncinata*.

The differential effects of warming on microarthropods and soil nematode abundance may be related to changing abiotic conditions in OTC chambers compared with control plots. Barcikowski and Loro [71] found species-specific differences in seasonal moss water content, showing that water content was both higher and less variable in *P. alpinum* when compared with *S. georgicouncinata*, further detailing how moss species can provide differential effects on the surrounding terrestrial environment. The availability of water and species-specific variation in moss water content and canopy temperatures we found with warming may influence both moss and soil fungi and microbial diversity, resulting in differential effects on microarthropod and soil nematode communities associated with bryophytes. In this study, we found that bryophyte fungal biomass was lower under passive warming, but also varied by species, which was contrary to our initial hypotheses. This decrease in moss-associated fungal biomass may be related to the significant increase in microarthropods associated with OTC warming, as both the moss-associated Oribatida and Collembola are known to feed on algae, dead organic material and fungi [72]. Clearly, more work is needed to better understand the effects of warming on bryophyte communities of the Western Antarctic Peninsula, including moss species-specific responses, both above and below ground, which may impact and alter the structure of invertebrate and microorganism communities and the function of this rapidly changing terrestrial ecosystem.

## 4.4. Species-specific effects, higher trophic levels and climate change

Research on bryophytes at higher latitudes suggests that mosses strongly influence ecosystem functions, including regulating water storage [21,25,73–75] and temperature [58,76]. In Antarctica, ecosystem function among bryophytes has primarily been assessed in terms of nutrient inputs [77], while little is known about how mosses may also influence the thermal and water regimes that impact ecosystem function. The work in northern latitudes indicate that bryophytes are species specific with respect to their impacts on ecosystem function, such as thermal properties [29,30], although little is known about the ecosystem effects of such differences.

Our results demonstrate that Antarctic bryophytes have species-specific effects in thermal absorption and soil water availability, which may directly affect the plant community and higher terrestrial trophic levels. We found moss species-specific responses to passive warming, showing differential canopy temperatures and shifts in invertebrate communities between the two dominant moss species, *P. alpinum* and *S. georgicouncinata*. Furthermore, we observed that warming increased sexual expression of *P. alpinum*, which was also linked to microarthropod abundance, illustrating the broad range of species-specific biological responses in terrestrial Antarctica bryophyte communities. While we found significant effects of warming and species in our experiments, we did not find significant interactions between passive warming and moss species. This suggests that at least initially during climate warming, the relative differences among species may remain constant. Thus, first and foremost, we must understand the functional effects of Antarctic plants on this changing ecosystem; however, our results indicate that there is a reason to be hopeful that the functional effects that we measure today will be predictive under future climate change scenarios. Many challenges remain to understanding the impacts of rapidly changing environmental conditions on terrestrial communities in Antarctica, and a major task is the still insufficient knowledge about terrestrial biodiversity and how this unique biodiversity (dominated by cryptogams, microorganisms and microfauna) links to larger ecosystem function in light of continued climate warming [11,18]. Microarthropods play a fundamental role in nutrient cycling within soil ecosystems, yet in the Antarctic terrestrial ecosystem these processes have been shown to be largely governed primarily by climatic parameters [68,78,79]. Future studies to quickly enhance our species-specific understanding of the responses of cryptogam communities to warming will be the key to develop an ecosystem scale understanding of terrestrial ecology and ecosystem function in a warming Antarctica.

Data accessibility. Data and code are available in the Dryad Digital Repository: https://doi.org/10.5061/dryad.mv32031 [80].
Authors' contributions. S.M.E., T.N.R., A.C.-K. and E.E.S. conceived the study and designed experiments. S.M.E., T.N.R., A.C.-K., H.M.P. and M.W.C were responsible for data collection in the field. H.M.P., A.F.C. and M.A.B. were responsible for data collection in the laboratory. S.M.E., H.M.P. and M.W.C. were responsible for data analysis and graphing. H.M.P. and S.M.E. were primarily responsible for the manuscript. All authors gave final approval for publication.
Competing interests. We have no competing interests.
Funding. This work was funded by the US National Science Foundation (NSF) (grant no. PLR 1341742 to S.M.E. and T.N.R.). A.C.-K. thanks the support of INACH (grant no. ECA 52-55) and the project INACH (grant no.

T0307), FONDECYT (grant no. 1181745). Thanks to the Chilean Antarctic Institute (INACH) for financial and outstanding logistical support.

Acknowledgements. Thanks to John Christy for identification of voucher samples. Further thanks to P. Zuniga, S. Kiel, S. Derkarabetian, C. Maraist, J. Shamek, S. Herrejon Chavez, E. Leal and T. Deakova for field, technical and laboratory support.

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
