## [Reviewer comments · Royal Society Open Science]

Review History

RSOS-190744.R0 (Original submission)

Review form: Reviewer 1 (Peter Convey)

Is the manuscript scientifically sound in its present form?

No

Are the interpretations and conclusions justified by the results?

No

Is the language acceptable?

Yes

Is it clear how to access all supporting data?

Yes

Do you have any ethical concerns with this paper?

No

Have you any concerns about statistical analyses in this paper?

Yes

Recommendation?

Major revision is needed (please make suggestions in comments)

Comments to the Author(s)

Species-specific effects of passive warming in an Antarctic moss system

Hannah M. Prather et al

This paper describes outcomes from a field environmental manipulation experiment carried out on the South Shetland Islands in the northern maritime Antarctic, attempting to place results in the context of potential impacts on the overall trophic web. Such longer-term passive experimental studies have been an important means used by researchers to attempt to model and predict the consequences of recent trends of environmental change in extreme terrestrial habitats such as those of the Antarctic and Arctic. An important element of this study is the attempt to look at species-specific outcomes of the manipulation. The ms is generally clearly written, though some editorial attention to syntax is required in places. My overall impression is that significant reduction could be made in the Discussion section, where several paragraphs/subsections are largely speculative and/or based on what are over-interpreted elements of the results obtained – similarly, the related Results sections could also be reduced.

Minor comments:

L26 – dominant, not dominate

L30-31 – as worded, this sentence for me implies some form of causal relationship between plant reproduction and invertebrates, which I can't see being the case, at least based on the evidence presented; rather, reproduction and vertebrates are both responding to the manipulation.

Intro para at L40 – introductions like this are common in many 'climate change' papers in the Antarctic literature, however they are currently slightly inaccurate. The situation described was valid up until around 2000, but since then there has been a well-documented 'pause' in the rapid warming in the Antarctic Peninsula region, with some indices currently even indicating cooling (see J Turner et al. 2016 Nature). This was also noted earlier by Parnikoza et al 2009 Global Change Biology, who documented a lack of further expansion of Argentine Islands higher plant populations in the 2000s. Warming and wetting trends (with different seasonal patterns) are also well documented from Signy Island (South Orkney Islands – see Royles et al 2012 GCB, and Cannone et al 2016 Climatic Change and 2017 Bot J Linn Soc) – perhaps an important relevant point is that summer precipitation both in these studies and in the S Shetlands is increasingly as rain, which is more immediately available to biota, and easier to measure (see also Convey 2011 Polar Biol).

L61 – another detail important to recognise is that the warming trends referred to have not affected 'all' of Antarctica. Most of the continent has shown no or very little trend in temperature, the strong trends being talked about have been restricted to the Antarctic Peninsula and Scotia Arc region (see SCAR ACCE report and update, Turner et al. 2009, 2014).

L66 – fauna include two species of insects (Diptera), both found on the S Shetlands.

L103 – be explicit here – how long has this experimental set-up been established?

L119 – given the overall richness (61 bryophytes) and the stated aim of this study to look at species-specific effects, in a way it is a little disappointing that the study is actually confined to

two species of moss! I accept that these are common species in this area and obviously at the study location itself, but neither are entirely 'representative' of maritime Antarctic mosses generally, and I think more thought is required than is currently apparent in the ms as to the limits to expanding from the current results to draw more general conclusions about change impacts on bryophytes generally.

L135 - again, to be explicit, are the OTCs installed year-round or (as with some studies of this type) removed overwinter to avoid excessive snow accumulation within them?

L150-152 - does this refer to the entire annual dataset, or was there also no overall effect on mean temps if the data were analysed seasonally?

L165 - given the intent of the study to look at plant-microenvironment-invertebrate interactions, it is a pity that temperatures were only recorded at 0.5 cm depth in the canopy, especially as many invertebrates are typically found for most of the time rather deeper in the canopy profile.

L168-170 - more detail required, and perhaps a diagrammatic illustration. In particular, how deep were the blocks, and where were the measurements made?

L184 - the core included the entire depth of the moss profile?

L220 - such small cores are likely to lead to low invertebrate numbers being obtained, but are obviously constrained by the overall size of the OTCs and the need not to compromise other aspects of this and other studies. I would suggest Mesostigmata should be separable as a group, as the only species in it present in this region is the predatory mite *Gamasellus racovitzai*, which is both large and distinctive. Indeed, given the aim of the study to draw inferences about trophic interactions, it would seem to be an advantage to try and separate out this taxon in the analyses.

L250-251 - surely to do this you also need to know what the initial cover values were at the start of the experiment? You can't assume that control and OTC plots were identical at that time. Same point applies at the Results interpretation at l323, and the entire paragraph at l326 - you can't compare 'present day' differences between treatment and control, if you don't know whether either has changed from the initial state at the start of the experiment.

L253 - vegetation, not vegetative (this comes up multiple times in the ms).

L277 - why not the other mite groups, were their numbers too low? In my experience, prostigmatid mites (which can be hard to locate even when alive) are generally much more abundant than oribatids in moss substrata, which leads to some concern as to whether the extraction carried out is actually effective/representative.

L282-284 - while the statement is true (limited to springtails I think, though am not sure; some citations would be useful) I believe the studies reporting it have been based on very different mosses to the Polytrichaceae moss studied here, while given that *Polytrichastrum* does not reproduce sexually in the Antarctic (extremely exceptionally), it seems unlikely to have an associated pollinator invertebrate in the Antarctic invertebrate fauna. As noted in comment on the Abstract I don't think evidence of a causal relationship can be inferred here, rather any correlation between responses is more likely to indicate separate responses to an underlying element of the environmental manipulation.

L300-301 - can't argue with the data, but this is a bit surprising when you think *Sanionia* is classically considered a hygic moss and found in often waterlogged ground, while *Polytrichastrum* is mesic and rarely if ever waterlogged.

L333 - were these two species present in these plots at the start of the experiment?

L341, 345 - It seems to be stating the obvious that the two mosses had different weight (and other morphological) characteristics - they are very different mosses. The primary interest in the results is whether or not differences/responses within species can be linked to the experimental manipulation.

L359 -this effect is marginally non-significant, not marginally significant.

L364-366 - these numbers emphasise my earlier concern about the potentially very small numbers of invertebrates obtained in this sampling methodology. Bear in mind there are quite a few papers based on extraction of larger cores that report densities in the 10's to 100's of thousands of individuals per square metre

L406 – in reality this is a very strong conclusion to draw based on the evidence obtained in this study – arguably the earlier study already referred to by some members of this group has documented a stronger example of this evidence. Similarly, given that the finding is really already known and published in other studies, the length of the para at L458 is more than is justified by the new data presented in this study.

Section 5.1 – while I agree with the overall thrust of what is said here, it has to be recognised that these are mostly general statements of what ‘could’ happen, and each represents quite a large logical ‘jump’ from the actual data obtained and presented here.

L440 – you can only say this, as noted above, if these species definitely were not present in the study plots at the start of the experiment. And even if not, given the very small number of instances, I can’t see that you can unequivocally tie the occurrence now to being caused by the experimental warming.

L448 – same comment, did it decrease relative to the original state in these plots?

L477 – as noted above, the subject in this paragraph (which is also more extended than it could be) is not really justified by the strength of the data presented in this study.

L498 – the paper that forms part of Day’s overall warming study reporting detailed analyses of responses of contained invertebrate communities is Convey et al. (2002, Ecology), which is referred to later in the same paragraph here. Nb that reference formatting in these paragraphs at this point in the ms differs from that elsewhere.

L512 – note that the Mouratov et al paper contains entirely incorrect identifications of the nematodes present on King George Island, erroneously based on a key from continental Antarctic Victoria Land. However, there are some very thorough studies of soil nematodes from Victoria Land by US researchers that document different soil water and salt ion preferences/tolerances of the nematode species that occur there.

Para at L517 – this another example of a long paragraph that in reality is largely speculative. In the absence of invertebrate identifications to species (or genus in nematodes; and note above comment about the fundamental error in identification and hence interpretation in Mouratov et al) it is virtually impossible to justify the sort of statements made in this paragraph, other than as unsubstantiated generalisations.

L558 – this is another overstatement of this point! The study very clearly did not observe any such change in reproductive output, as no sporophytes were observed at all!

References – note miss-spelling of Bokhorst at points (not Bokhurst). There are clear formatting inconsistencies and errors throughout the ref list.

Review form: Reviewer 2 (Filipe Victoria)

Is the manuscript scientifically sound in its present form?

No

Are the interpretations and conclusions justified by the results?

No

Is the language acceptable?

Yes

Is it clear how to access all supporting data?

No

Do you have any ethical concerns with this paper?

No

Have you any concerns about statistical analyses in this paper?

I do not feel qualified to assess the statistics

Recommendation?

Major revision is needed (please make suggestions in comments)

Comments to the Author(s)

This paper draws attention by suggesting data that may demonstrate the behavior of Antarctic plants in the face of environmental changes, especially warming in Antarctica.

However, the information presented is still very superficial, only two species were effectively evaluated. Therefore, see tem OTCs in several others sites in King George Island, why don't include the data of these? Why other patches of different species were no evaluate also? Based all the discussion on these couple species could be configure an over extrapolation of the results. The reproductive biology methods does not sound right to do. The age of moss caulonemata its too difficult to define only by colours. And as well, as an extremely dry environment, seek for sporophytes is quite be an challenge, mainly the female gamethophores will be found frequently. Its is an major issue that could unworthy the paper results.

The section with the most security in the presentation of results and discussions is the one that deals with comparing the anthropofauna associated with the moss spots. There I believe they have a more robust and exploitable result.

I suggest to re-write the paper concerning only in such founds.

Decision letter (RSOS-190744.R0)

08-Aug-2019

Dear Dr Prather,

The editors assigned to your paper ("Species-specific effects of passive warming in an Antarctic moss system") have now received comments from reviewers. We would like you to revise your paper in accordance with the referee and Associate Editor suggestions which can be found below (not including confidential reports to the Editor). Please note this decision does not guarantee eventual acceptance.

Please submit a copy of your revised paper before 31-Aug-2019. Please note that the revision deadline will expire at 00.00am on this date. If we do not hear from you within this time then it will be assumed that the paper has been withdrawn. In exceptional circumstances, extensions may be possible if agreed with the Editorial Office in advance. We do not allow multiple rounds of revision so we urge you to make every effort to fully address all of the comments at this stage. If deemed necessary by the Editors, your manuscript will be sent back to one or more of the original reviewers for assessment. If the original reviewers are not available, we may invite new reviewers.

- Data accessibility

If you wish to submit your supporting data or code to Dryad (<http://datadryad.org/>), or modify your current submission to dryad, please use the following link:
<http://datadryad.org/submit?journalID=RSOS&manu=RSOS-190744>

- Competing interests

- Authors' contributions

- Acknowledgements

- Funding statement

Kind regards,

on behalf of Professor Kevin Padian (Subject Editor)
openscience@royalsociety.org

Associate Editor's comments to the Author:

The reviewers of your paper have made a number of important recommendations that you will need to address in any revision (and to which you will need to provide a point-by-point response). Please ensure you make every effort to resolve their concerns, as the journal can only provide one round of major revision in general. Thanks for the submission and good luck!

Reviewers' Comments to Author:

Reviewer: 1
Comments to the Author(s)

Species-specific effects of passive warming in an Antarctic moss system

Hannah M. Prather et al

This paper describes outcomes from a field environmental manipulation experiment carried out on the South Shetland Islands in the northern maritime Antarctic, attempting to place results in the context of potential impacts on the overall trophic web. Such longer-term passive experimental studies have been an important means used by researchers to attempt to model and predict the consequences of recent trends of environmental change in extreme terrestrial habitats such as those of the Antarctic and Arctic. An important element of this study is the attempt to look at species-specific outcomes of the manipulation. The ms is generally clearly written, though some editorial attention to syntax is required in places. My overall impression is that significant reduction could be made in the Discussion section, where several paragraphs/subsections are largely speculative and/or based on what are over-interpreted elements of the results obtained – similarly, the related Results sections could also be reduced.

Minor comments:

L26 – dominant, not dominate

L30-31 – as worded, this sentence for me implies some form of causal relationship between plant reproduction and invertebrates, which I can't see being the case, at least based on the evidence presented; rather, reproduction and vertebrates are both responding to the manipulation.

Intro para at L40 – introductions like this are common in many 'climate change' papers in the Antarctic literature, however they are currently slightly inaccurate. The situation described was valid up until around 2000, but since then there has been a well-documented 'pause' in the rapid warming in the Antarctic Peninsula region, with some indices currently even indicating cooling (see J Turner et al. 2016 Nature). This was also noted earlier by Parnikoza et al 2009 Global Change Biology, who documented a lack of further expansion of Argentine Islands higher plant populations in the 2000s. Warming and wetting trends (with different seasonal patterns) are also well documented from Signy Island (South Orkney Islands – see Royles et al 2012 GCB, and Cannone et al 2016 Climatic Change and 2017 Bot J Linn Soc) – perhaps an important relevant point is that summer precipitation both in these studies and in the S Shetlands is increasingly as rain, which is more immediately available to biota, and easier to measure (see also Convey 2011 Polar Biol).

L61 – another detail important to recognise is that the warming trends referred to have not affected 'all' of Antarctica. Most of the continent has shown no or very little trend in temperature, the strong trends being talked about have been restricted to the Antarctic Peninsula and Scotia Arc region (see SCAR ACCE report and update, Turner et al. 2009, 2014).

L66 – fauna include two species of insects (Diptera), both found on the S Shetlands.

L103 – be explicit here – how long has this experimental set-up been established?

L119 – given the overall richness (61 bryophytes) and the stated aim of this study to look at species-specific effects, in a way it is a little disappointing that the study is actually confined to two species of moss! I accept that these are common species in this area and obviously at the study location itself, but neither are entirely 'representative' of maritime Antarctic mosses generally, and I think more thought is required than is currently apparent in the ms as to the limits to expanding from the current results to draw more general conclusions about change impacts on bryophytes generally.

L135 – again, to be explicit, are the OTCs installed year-round or (as with some studies of this type) removed overwinter to avoid excessive snow accumulation within them?

L150-152 – does this refer to the entire annual dataset, or was there also no overall effect on mean temps if the data were analysed seasonally?

L165 – given the intent of the study to look at plant-microenvironment-invertebrate interactions, it is a pity that temperatures were only recorded at 0.5 cm depth in the canopy, especially as many invertebrates are typically found for most of the time rather deeper in the canopy profile.

L168-170 – more detail required, and perhaps a diagrammatic illustration. In particular, how deep were the blocks, and where were the measurements made?

L184 – the core included the entire depth of the moss profile?

L220 – such small cores are likely to lead to low invertebrate numbers being obtained, but are obviously constrained by the overall size of the OTCs and the need not to compromise other aspects of this and other studies. I would suggest Mesostigmata should be separable as a group, as the only species in this region is the predatory mite *Gamasellus racovitzai*, which is both large and distinctive. Indeed, given the aim of the study to draw inferences about trophic interactions, it would seem to be an advantage to try and separate out this taxon in the analyses.

L250-251 – surely to do this you also need to know what the initial cover values were at the start of the experiment? You can't assume that control and OTC plots were identical at that time. Same point applies at the Results interpretation at l323, and the entire paragraph at l326 – you can't compare 'present day' differences between treatment and control, if you don't know whether either has changed from the initial state at the start of the experiment.

L253 – vegetation, not vegetative (this comes up multiple times in the ms).

L277 – why not the other mite groups, were their numbers too low? In my experience, prostigmatid mites (which can be hard to locate even when alive) are generally much more abundant than oribatids in moss substrata, which leads to some concern as to whether the extraction carried out is actually effective/representative.

L282-284 – while the statement is true (limited to springtails I think, though am not sure; some citations would be useful) I believe the studies reporting it have been based on very different mosses to the Polytrichaceae moss studied here, while given that Polytrichastrum does not reproduce sexually in the Antarctic (extremely exceptionally), it seems unlikely to have an associated pollinator invertebrate in the Antarctic invertebrate fauna. As noted in comment on the Abstract I don't think evidence of a causal relationship can be inferred here, rather any correlation between responses is more likely to indicate separate responses to an underlying element of the environmental manipulation.

L300-301 – can't argue with the data, but this is a bit surprising when you think Sanionia is classically considered a hygic moss and found in often waterlogged ground, while Polytrichastrum is mesic and rarely if ever waterlogged.

L333 – were these two species present in these plots at the start of the experiment?

L341, 345 – It seems to be stating the obvious that the two mosses had different weight (and other morphological) characteristics – they are very different mosses. The primary interest in the results is whether or not differences/responses within species can be linked to the experimental manipulation.

L359 – this effect is marginally non-significant, not marginally significant.

L364-366 – these numbers emphasise my earlier concern about the potentially very small numbers of invertebrates obtained in this sampling methodology. Bear in mind there are quite a few papers based on extraction of larger cores that report densities in the 10's to 100's of thousands of individuals per square metre

L406 – in reality this is a very strong conclusion to draw based on the evidence obtained in this study – arguably the earlier study already referred to by some members of this group has documented a stronger example of this evidence. Similarly, given that the finding is really already known and published in other studies, the length of the para at L458 is more than is justified by the new data presented in this study.

Section 5.1 – while I agree with the overall thrust of what is said here, it has to be recognised that these are mostly general statements of what 'could' happen, and each represents quite a large logical 'jump' from the actual data obtained and presented here.

L440 – you can only say this, as noted above, if these species definitely were not present in the study plots at the start of the experiment. And even if not, given the very small number of instances, I can't see that you can unequivocally tie the occurrence now to being caused by the experimental warming.

L448 – same comment, did it decrease relative to the original state in these plots?

L477 – as noted above, the subject in this paragraph (which is also more extended than it could be) is not really justified by the strength of the data presented in this study.

L498 – the paper that forms part of Day's overall warming study reporting detailed analyses of responses of contained invertebrate communities is Convey et al. (2002, Ecology), which is referred to later in the same paragraph here. Nb that reference formatting in these paragraphs at this point in the ms differs from that elsewhere.

L512 – note that the Mouratov et al paper contains entirely incorrect identifications of the nematodes present on King George Island, erroneously based on a key from continental Antarctic Victoria Land. However, there are some very thorough studies of soil nematodes from Victoria Land by US researchers that document different soil water and salt ion preferences/tolerances of the nematode species that occur there.

Para at L517 – this another example of a long paragraph that in reality is largely speculative. In the absence of invertebrate identifications to species (or genus in nematodes; and note above comment about the fundamental error in identification and hence interpretation in Mouratov et

al) it is virtually impossible to justify the sort of statements made in this paragraph, other than as unsubstantiated generalisations.

L558 - this is another overstatement of this point! The study very clearly did not observe any such change in reproductive output, as no sporophytes were observed at all!

References - note miss-spelling of Bokhorst at points (not Bokhurst). There are clear formatting inconsistencies and errors throughout the ref list.

Reviewer: 2

Comments to the Author(s)

This paper draws attention by suggesting data that may demonstrate the behavior of Antarctic plants in the face of environmental changes, especially warming in Antarctica.

However, the information presented is still very superficial, only two species were effectively evaluated. Therefore, see tem OTCs in several others sites in King George Island, why don't include the data of these? Why other patches of different species were no evaluate also? Based all the discussion on these couple species could be configure an over extrapolation of the results.

The reproductive biology methods does not sound right to do. The age of moss caulonemata its too difficult to define only by colours. And as well, as an extremely dry environment, seek for sporophytes is quite be an challenge, mainly the female gamethophores will be found frequently. Its is an major issue that could unworthy the paper results.

The section with the most security in the presentation of results and discussions is the one that deals with comparing the anthropofauna associated with the moss spots. There I believe they have a more robust and exploitable result.

I suggest to re-write the paper concerning only in such founds.

Author's Response to Decision Letter for (RSOS-190744.R0)

See Appendix A.

RSOS-190744.R1 (Revision)

Review form: Reviewer 1 (Peter Convey)

Is the manuscript scientifically sound in its present form?

Yes

Are the interpretations and conclusions justified by the results?

Yes

Is the language acceptable?

Yes

Do you have any ethical concerns with this paper?

No

Have you any concerns about statistical analyses in this paper?

No

Recommendation?

Accept with minor revision (please list in comments)

Comments to the Author(s)

Species-specific effects of passive warming in an Antarctic moss system

Hannah M. Prather et al (revision)

The authors have made good efforts to respond to the matters raised by both reviewers, and I am satisfied with the responses and overall changes. Reading the new version of the ms raises the following minor issues:

L19 - delete 'regions' - the sentence doesn't read correctly as it is currently

L26 - 'dominant' not 'dominate'

L29-31 - the concept of 'reproductive shifts' and particularly 'reproduction' is still stated too strongly here - there was no change in actual reproduction in the expt (ie no sporophytes), and this idea needs to be de-emphasised.

L34 - what does 'differentially impact' mean in the context of this sentence? I think I would reduce this to 'may impact...'

L55 - 'Signy' is miss-spelt

L100 - 'Antarctic'

L101-4 - the statement about the current lack of these studies is correct but, again, in that the current study did not find any form of change in actual reproduction or in reproductive output, in essence this is a subject that the current study cannot actually address - I do not think it is appropriate to say, in effect, 'if we had found any changes then this study would have addressed this subject'!

L106 - what is 'Antarctic terrestrialization'?

L121 - archipelago should not be capitalised

L123 - delete 'unique' - any confirmed species a unique species in some sense.

L132 - 'sexually reproductive' - plainly they frequently reproduce asexually

L170 - 'recordings were made'

L280 - oribatid should not be capitalised, also at l346 and l358 (the proper name of the group, Oribatida, would be)

L283 - 'and the number'

L298-300 - it seems odd that all three ANOVAs have the same F value here, especially as there are two different p values as well but identical sample sizes (and again at l303/4)?

L386 - 'higher trophic levels' - some caution needed here and elsewhere in the ms, as some readers will take this as implying that the mites, springtails or nematodes actively graze on the mosses (producers), which they do not. Rather they are microbivores - if they graze on living algae (likely for the springtails) then they can be considered primary consumers. But many will rather be detritivores, grazing decaying dead plant material and the bacterial/fungal components therein. There are certainly not plural higher trophic levels examined in this study.

L394 - control

L439 – picking up on one of the review responses, the title of the Day et al 2009 is indeed as referred, but the primary research paper reporting the Collembola study which it summarises, and which took place as part of the Day group's overall project, is that of Convey et al. (2002).

L441 – capitalise 'Peninsula'

L444 – delete 'endemic' and capitalise 'Collembola'

L470 – capitalise 'Oribatida' and 'Collembola'; also the general statement that these microarthropods are fungivores is an overstatement of the information the cited reference gives. In fact, many springtails and in particular the commonly dominant *Cryptopygus antarcticus* are algivores rather than fungivores (see also earlier comment on whether these species can be more appropriately be considered in the decomposition cycle rather than as 'higher trophic levels').

L514 – Chilean Antarctic Institute

Review form: Reviewer 2 (Filipe Victoria)

Is the manuscript scientifically sound in its present form?

Yes

Are the interpretations and conclusions justified by the results?

Yes

Is the language acceptable?

Yes

Do you have any ethical concerns with this paper?

Yes

Have you any concerns about statistical analyses in this paper?

No

Recommendation?

Accept with minor revision (please list in comments)

Comments to the Author(s)

Re-reading the manuscript sent by the authors, it is noted that an extensive improvement in the arguments was made, giving greater reliability to the results. However, I believe that it is necessary to give due clarity to the experiment. The authors argue that they used the usual methodology for specimen identification and reproductive organ evaluation in *P. alpinum* samples. In other words, they had access to physical material for gametophyte phenotyping analysis. What I would like to see, and would give more transparency to the data, would be images of these gametophytes, in their habit, to illustrate to readers that they are even these two species (mainly for *S. georgicouncinata* which is strongly disputed from the point of view). species circumscription), as well as images of the archegons identified in the sexual expression analyzes.

On the other hand, would you like to know if male individuals were found outside OTCs? Those who explore and research the Antarctic vegetation, especially those that go on for more than 20 years annually, report no sporophytes of some species, and polytrichaceas are among them (if I'm not mistaken the last finding was in the 1990s, at work). published by Peter Convey and Prof. R. Lewis-Smith). Much is still theorized about the advantage of vegetative rather than sexual reproduction for these species because of the extreme climate. But it can also be thought of because there's not so many individuals representing both sexes in the same compartment (In Admiralty Bay, for example, it is recurrent to find male individuals with splash-cups, and few or

almost no female individuals), or about decreased water availability in areas where vegetation grows in Antarctica, so I would like to see a sign of this at work, if only to show evidence that male individuals were found outside the OTC, including images if they are still available. .

Regarding the modifications requested by the reviewer I, I believe that this one will also manifest, but I think it still needs to tie up the different associated experiments. I can understand all the reasons that led the authors to perform these experiments, but I had the happy opportunity to read and follow two different moments in the development of this work, and the reader who will access the final article after the acceptance will not have this same opportunity. .

So I believe there is still some information (unfortunate in the introduction) that can tie the issue of heating, with the sex ratio and the arthropods.

Decision letter (RSOS-190744.R1)

28-Sep-2019

Dear Dr Prather:

On behalf of the Editors, I am pleased to inform you that your Manuscript RSOS-190744.R1 entitled "Species-specific effects of passive warming in an Antarctic moss system" has been accepted for publication in Royal Society Open Science subject to minor revision in accordance with the referee suggestions. Please find the referees' comments at the end of this email.

The reviewers and Subject Editor have recommended publication, but also suggest some minor revisions to your manuscript. Therefore, I invite you to respond to the comments and revise your manuscript.

- Ethics statement

- Data accessibility

If you wish to submit your supporting data or code to Dryad (<http://datadryad.org/>), or modify your current submission to dryad, please use the following link:
<http://datadryad.org/submit?journalID=RSOS&manu=RSOS-190744.R1>

- Competing interests

- Authors' contributions

- Acknowledgements

- Funding statement

Because the schedule for publication is very tight, it is a condition of publication that you submit the revised version of your manuscript before 07-Oct-2019. Please note that the revision deadline will expire at 00.00am on this date. If you do not think you will be able to meet this date please let me know immediately.

- 1) A text file of the manuscript (tex, txt, rtf, docx or doc), references, tables (including captions) and figure captions. Do not upload a PDF as your "Main Document".
- 2) A separate electronic file of each figure (EPS or print-quality PDF preferred (either format should be produced directly from original creation package), or original software format)

- 3) Included a 100 word media summary of your paper when requested at submission. Please ensure you have entered correct contact details (email, institution and telephone) in your user account
- 4) Included the raw data to support the claims made in your paper. You can either include your data as electronic supplementary material or upload to a repository and include the relevant doi within your manuscript
- 5) All supplementary materials accompanying an accepted article will be treated as in their final form. Note that the Royal Society will neither edit nor typeset supplementary material and it will be hosted as provided. Please ensure that the supplementary material includes the paper details where possible (authors, article title, journal name).

on behalf of Prof Kevin Padian (Subject Editor)
openscience@royalsociety.org

Associate Editor Comments to Author:

The reviewers and editors are grateful for the efforts you've made to improve the manuscript; however, they identify a number of changes necessary to get the paper over the line. Please carefully respond to the reviewers' recommendations before you resubmit.

Reviewer comments to Author:

Reviewer: 2

Comments to the Author(s)

Re-reading the manuscript sent by the authors, it is noted that an extensive improvement in the arguments was made, giving greater reliability to the results. However, I believe that it is necessary to give due clarity to the experiment. The authors argue that they used the usual methodology for specimen identification and reproductive organ evaluation in *P. alpinum* samples. In other words, they had access to physical material for gametophyte phenotyping analysis. What I would like to see, and would give more transparency to the data, would be images of these gametophytes, in their habit, to illustrate to readers that they are even these two species (mainly for *S. georgicouncinata* which is strongly disputed from the point of view). species circumscription), as well as images of the archegons identified in the sexual expression analyzes.

On the other hand, would you like to know if male individuals were found outside OTCs? Those who explore and research the Antarctic vegetation, especially those that go on for more than 20 years annually, report no sporophytes of some species, and polytrichaceas are among them (if I'm not mistaken the last finding was in the 1990s, at work). published by Peter Convey and Prof. R. Lewis-Smith). Much is still theorized about the advantage of vegetative rather than sexual reproduction for these species because of the extreme climate. But it can also be thought of because there's not so many individuals representing both sexes in the same compartment (In Admiralty Bay, for example, it is recurrent to find male individuals with splash-cups, and few or almost no female individuals), or about decreased water availability in areas where vegetation grows in Antarctica, so I would like to see a sign of this at work, if only to show evidence that male individuals were found outside the OTC, including images if they are still available. .

Regarding the modifications requested by the reviewer I, I believe that this one will also manifest, but I think it still needs to tie up the different associated experiments. I can understand all the reasons that led the authors to perform these experiments, but I had the happy opportunity to read and follow two different moments in the development of this work, and the reader who will access the final article after the acceptance will not have this same opportunity. .
So I believe there is still some information (unfortunate in the introduction) that can tie the issue of heating, with the sex ratio and the arthropods.

Reviewer: 1

Comments to the Author(s)
Royal Society Open Science

Species-specific effects of passive warming in an Antarctic moss system

Hannah M. Prather et al (revision)

The authors have made good efforts to respond to the matters raised by both reviewers, and I am satisfied with the responses and overall changes. Reading the new version of the ms raises the following minor issues:

L19 - delete 'regions' - the sentence doesn't read correctly as it is currently

L26 - 'dominant' not 'dominate'

L29-31 - the concept of 'reproductive shifts' and particularly 'reproduction' is still stated too strongly here - there was no change in actual reproduction in the expt (ie no sporophytes), and this idea needs to be de-emphasised.

L34 - what does 'differentially impact' mean in the context of this sentence? I think I would reduce this to 'may impact...'

L55 - 'Signy' is miss-spelt

L100 - 'Antarctic'

L101-4 - the statement about the current lack of these studies is correct but, again, in that the current study did not find any form of change in actual reproduction or in reproductive output, in essence this is a subject that the current study cannot actually address - I do not think it is appropriate to say, in effect, 'if we had found any changes then this study would have addressed this subject'!

L106 - what is 'Antarctic terrestrialization'?

L121 - archipelago should not be capitalised

L123 - delete 'unique' - any confirmed species a unique species in some sense.

L132 - 'sexually reproductive' - plainly they frequently reproduce asexually

L170 - 'recordings were made'

L280 - oribatid should not be capitalised, also at l346 and l358 (the proper name of the group, Oribatida, would be)

L283 - 'and the number'

L298-300 - it seems odd that all three ANOVAs have the same F value here, especially as there are two different p values as well but identical sample sizes (and again at l303/4)?

L386 - 'higher trophic levels' - some caution needed here and elsewhere in the ms, as some readers will take this as implying that the mites, springtails or nematodes actively graze on the mosses (producers), which they do not. Rather they are microbivores - if they graze on living algae (likely for the springtails) then they can be considered primary consumers. But many will rather be detritivores, grazing decaying dead plant material and the bacterial/fungal components therein. There are certainly not plural higher trophic levels examined in this study.

L394 - control

L439 - picking up on one of the review responses, the title of the Day et al 2009 is indeed as referred, but the primary research paper reporting the Collembola study which it summarises, and which took place as part of the Day group's overall project, is that of Convey et al. (2002).

L441 - capitalise 'Peninsula'

L444 - delete 'endemic' and capitalise 'Collembola'

L470 - capitalise 'Oribatida' and 'Collembola'; also the general statement that these microarthropods are fungivores is an overstatement of the information the cited reference gives. In fact, many springtails and in particular the commonly dominant *Cryptopygus antarcticus* are algivores rather than fungivores (see also earlier comment on whether these species can be more appropriately be considered in the decomposition cycle rather than as 'higher trophic levels').

L514 - Chilean Antarctic Institute

Author's Response to Decision Letter for (RSOS-190744.R1)

See Appendix B.

Decision letter (RSOS-190744.R2)

09-Oct-2019

Dear Dr Prather,

I am pleased to inform you that your manuscript entitled "Species-specific effects of passive warming in an Antarctic moss system" is now accepted for publication in Royal Society Open Science.

Kind regards,
Anita Kristiansen
Editorial Coordinator
Royal Society Open Science
openscience@royalsociety.org

Kevin Padian (Subject Editor)
openscience@royalsociety.org

Appendix A

27 August 2019

Dear Professor Kevin Padian
Open Science, Subject Editor

Dear Dr. Padian:

We are resubmitting our manuscript "Species-specific effects of passive warming in an Antarctic moss system" (RSOS ID 190744) for publication in *Open Science*. Our manuscript provides novel data on the effects of warming on Antarctic plant communities across trophic levels.

In this revised manuscript, you will find that we have carefully addressed the reviewers' comments. In particular, we made the Results and Discussion sections more concise, as suggested. We have made other minor changes suggested, and we have indicated our point-by-point responses below (in italics).

Thank you for your patience, time and consideration, and we appreciate the careful consideration of our manuscript by the reviewers, which has greatly improved it. We look forward to hearing from you at your earliest convenience.

Sincerely,
Hannah Prather

Associate Editor's comments to the Author:

The reviewers of your paper have made a number of important recommendations that you will need to address in any revision (and to which you will need to provide a point-by-point response). Please ensure you make every effort to resolve their concerns, as the journal can only provide one round of major revision in general. Thanks for the submission and good luck!

Reviewers' Comments to Author:

**Reviewer: 1
Comments to the Author(s)**

Species-specific effects of passive warming in an Antarctic moss system

Hannah M. Prather et al

This paper describes outcomes from a field environmental manipulation experiment carried out on the South Shetland Islands in the northern maritime Antarctic, attempting to place results in the context of potential impacts on the overall trophic web. Such longer-term passive experimental studies have been an important means used by researchers to attempt to model and predict the consequences of recent trends of environmental change in extreme terrestrial habitats such as those of the Antarctic and Arctic. An important element of this study is the attempt to look at species-specific outcomes of the manipulation. The ms is generally clearly written, though some editorial attention to syntax is required in places. My overall impression is that significant reduction could be made in the Discussion section, where several paragraphs/subsections are largely speculative and/or based on what are over-interpreted elements of the results obtained – similarly, the related Results sections could also be reduced.

*As suggested by Reviewer 1, we have made the Results and Discussion section more concise and made the Discussion less speculative. Specifically, we made major changes to Results section 4.2 (Cryptogam community response to passive warming), Results section 4.4 (Sex expression in *P. alpinum*), Discussion section 5.1 (Species-specific thermal effects observed in bryophytes under experimental warming), Discussion section 5.2 (Effects of experimental warming on moss species), and Discussion section 5.3 (Warming and moss species differentially affects higher trophic levels).*

Minor comments:

L26 – dominant, not dominate

Current L26. This has been corrected.

L30-31 – as worded, this sentence for me implies some form of causal relationship between plant reproduction and invertebrates, which I can't see being the case, at least based on the evidence presented; rather, reproduction and vertebrates are both responding to the manipulation.

Current L30-31. We have used the word “correlation” to be clear we do not intend to imply causation, and we now included a phrase to clarify that we do not know the underlying cause of this correlation.

Intro para at L40 – introductions like this are common in many ‘climate change’ papers in the Antarctic literature, however they are currently slightly inaccurate. The situation described was valid up until around 2000, but since then there has been a well-documented ‘pause’ in the rapid warming in the Antarctic Peninsula region, with some indices currently even indicating cooling (see J Turner et al. 2016 Nature). This was also noted earlier by Parnikoza et al 2009 Global Change Biology, who documented a lack of further expansion of Argentine Islands higher plant populations in the 2000s.

Current L40-57. We have explained the more nuanced changes to climate in Polar regions and in the region in which we worked (citing Turner et al. 2016).

Warming and wetting trends (with different seasonal patterns) are also well documented from Signy Island (South Orkney Islands – see Royles et al 2012 GCB, and Cannone et al 2016 Climatic Change and 2017 Bot J Linn Soc) – perhaps an important relevant point is that summer precipitation both in these studies and in the S Shetlands is increasingly as rain, which is more immediately available to biota, and easier to measure (see also Convey 2011 Polar Biol).

Current L52-59. As suggested, we included more information about the wetting trend in the South Shetland Islands, where we conducted our research. We expand on our explanation of the relationship between this trend and biodiversity.

L61 – another detail important to recognise is that the warming trends referred to have not affected ‘all’ of Antarctica. Most of the continent has shown no or very little trend in temperature, the strong trends being talked about have been restricted to the Antarctic Peninsula and Scotia Arc region (see SCAR ACCE report and update, Turner et al. 2009, 2014).

Current L64. We have clarified that we are talking about certain regions of Antarctica. We agree that we need to be careful not to overgeneralize.

L66 – fauna include two species of insects (Diptera), both found on the S Shetlands.

Current L69. We have included this information.

L103 – be explicit here – how long has this experimental set-up been established?

Current L108. We have added the information in the Introduction that that the experiment lasted 8 years (2008-2016). We agree that it is better to include this information here, as well as in the Methods.

L119 – given the overall richness (61 bryophytes) and the stated aim of this study to look at species-specific effects, in a way it is a little disappointing that the study is actually confined to two species of moss! I accept that these are common species in this area and obviously at the study location itself, but neither are entirely ‘representative’ of maritime Antarctic mosses generally, and I think more thought is required than is currently apparent in the ms as to the limits to expanding from the current results to draw more general conclusions about change impacts on bryophytes generally.

As we mentioned above, we have made our Discussion less speculative.

L135 – again, to be explicit, are the OTCs installed year-round or (as with some studies of this type) removed overwinter to avoid excessive snow accumulation within them?

Current L148-149. We have made it clear that they are installed year-round.

L150-152 – does this refer to the entire annual dataset, or was there also no overall effect on mean temps if the data were analysed seasonally?

Current L155-156. We have clarified this point. This referred to the entire annual dataset, and there was no effect on mean temperatures for the annual dataset or when analyzed seasonally.

L165 – given the intent of the study to look at plant-microenvironment-invertebrate interactions, it is a pity that temperatures were only recorded at 0.5 cm depth in the canopy, especially as many invertebrates are typically found for most of the time rather deeper in the canopy profile.

We agree that more information is needed on heat transfer across moss canopies, and we are currently collecting this data for a wide range of Antarctic moss species. However, these data are an excellent first examination of how canopy temperature in individual moss species may affect invertebrates.

L168-170 – more detail required, and perhaps a diagrammatic illustration. In particular, how deep were the blocks, and where were the measurements made?

Current L173-176. We have more clearly explained the depth of these moss blocks and how the measurements were made.

L184 – the core included the entire depth of the moss profile?

Current L189-190. We have clarified that the cores included the entire moss profile from rhizoids to the tops of the plants.

L220 – such small cores are likely to lead to low invertebrate numbers being obtained, but are obviously constrained by the overall size of the OTCs and the need not to compromise other aspects of this and other studies. I would suggest Mesostigmata should be separable as a group, as the only species in it present in this region is the predatory mite Gamasellus racovitzai, which is both large and distinctive. Indeed, given the aim of the study to draw inferences about trophic interactions, it would seem to be an advantage to try and separate out this taxon in the analyses.

We have clarified this point. Our invertebrate results, abundance and taxonomic rank, are comparable to those found in other invertebrate studies on the Western Antarctic Peninsula, (see Bokhorst et al. 2016 Polar Biology and Bokhorst & Convey 2016 Antarctic Science). Gamasellus racovitzai was not detected in this study, which is likely due to the extreme site variation across and within the islands of maritime Antarctica. We are confident that our extractions and identifications have been performed carefully and using common methodology under the training of Entomologist Dr. Moldenke at Oregon State University.

L250-251 – surely to do this you also need to know what the initial cover values were are the start of the experiment? You can't assume that control and OTC plots were identical at that time. Same point applies at the Results interpretation at I323, and the entire paragraph at I326 – you can't compare 'present day' differences between treatment and control, if you don't know whether either has changed from the initial state at the start of the experiment.

Current L256, 306-315. We have removed our analyses of species richness, total plant percent cover, and the bryophyte percent cover differences between warmed and control treatments. We have also removed Table 1. In particular, we agree that individual analysis of species needs initial cover values for each species. We have retained our descriptions on the bryophyte species in each treatment, and the NMDS of cryptogam communities, in which we found no significant differences between treatments. This is similar to what we have done in a previously published study from long term OTCs at a nearby site on King George Island (Shortlidge et al. 2017 Annals of Botany). Because our treatments were installed for eight years, with control and warming treatment paired by micro-site, we feel that this analysis examining overall community similarity is robust.

L253 – vegetation, not vegetative (this comes up multiple times in the ms).

Current L256. We have made this change here and throughout the manuscript.

L277 – why not the other mite groups, were their numbers too low? In my experience, prostigmatid mites (which can be hard to locate even when alive) are generally much more abundant than oribatids in moss substrata, which leads to some concern as to whether the extraction carried out is actually effective/representative.

We were extremely thorough with our extraction methods and have been extracting invertebrates from temperate mosses for many years. We feel we are able to determine the reliability of the method. Additionally, at other nearby sites, we have extracted more mites at different ratios, suggesting to us that this site is probably dictating the number and nature of mites, rather than the extraction method. Due to extreme microsite differences in moisture and temperature, we are not surprised to see such differences. It would be better if we could have carried this out this experiment at more sites, but logistics limit that option in this case.

L282-284 – while the statement is true (limited to springtails I think, though am not sure; some citations would be useful) I believe the studies reporting it have been based on very different mosses to the Polytrichaceae moss studied here, while given that Polytrichastrum does not reproduce sexually in the Antarctic (extremely exceptionally), it seems unlikely to have an associated pollinator invertebrate in the Antarctic invertebrate fauna. As noted in comment on the Abstract I don't think evidence of a causal relationship can be inferred here, rather any correlation between responses is more likely to indicate separate responses to an underlying element of the environmental manipulation.

Current L285. We have replaced “influenced by” with “correlated with”.

Current L415-432. We moved our discussion of the reasoning for and interpretation of this analysis to the Discussion, so that we do not repeat the information. Mites and springtails of a variety of species have been found to increase sexual reproduction in several moss species in several moss families, and this mode of fertilization is widely accepted by moss reproductive biologists internationally, although the relative importance of biotic (microarthropods) and abiotic (water) in moss fertilization needs much more research. The studies so far have shown that there is no specialization (no need to have evolved specialized microarthropods), but that mites and springtails increase sexual reproduction in mosses synergistically with water (i.e. that sperm are being transported by both water and microarthropods. Also, microarthropods, favor sexual expressing moss shoots versus non-expressing shoots, potentially altering trophic dynamics as plant reproduction shifts.

L300-301 – can't argue with the data, but this is a bit surprising when you think Sanionia is classically considered a hygic moss and found in often waterlogged ground, while Polytrichastrum is mesic and rarely if ever waterlogged.

We agree that this is not what we expect to find given the character of these two species.

L333 – were these two species present in these plots at the start of the experiment?

These species occur at very low frequency at this site. Therefore, it is possible that they occurred in other plots. We have removed any mention of this topic from the Discussion.

L341, 345 – It seems to be stating the obvious that the two mosses had different weight (and other morphological) characteristics – they are very different mosses. The primary interest in the results is whether or not differences/responses within species can be linked to the experimental manipulation.

We agree that moss species will be different (although some studies lump them together as mosses), but we know little about how these differences affect invertebrates (with or without warming).

L359 –this effect is marginally non-significant, not marginally significant.

Current L340-341. We have rephrased this.

L364-366 – these numbers emphasise my earlier concern about the potentially very small numbers of invertebrates obtained in this sampling methodology. Bear in mind there are quite a few papers based on extraction of larger cores that report densities in the 10's to 100's of thousands of individuals per square metre

We have clarified this point. As noted above, we find that our results are comparable to other studies along the Western Antarctica Peninsula and acknowledge the extreme amount of local site variation that exists in the Maritime Antarctic ecosystem. Further, we are confident that our extractions and identifications have been performed carefully and using common methodology under the training of Entomologist Dr. Moldenke at Oregon State University.

L406 – in reality this is a very strong conclusion to draw based on the evidence obtained in this study – arguably the earlier study already referred to by some members of this group has documented a stronger example of this evidence. Similarly, given that the finding is really already known and published in other studies, the length of the para at L458 is more than is justified by the new data presented in this study.

Current L387. We agree that we need to restate our conclusion. In this study, we showed that warming will influence “moss sexual expression” not necessarily “sexual reproduction” to production of spores, and we have made this clearer. We have made this section 5.2 (including the paragraph which originally included L458) much shorter and combined it into one paragraph.

Current L421-432. We have made this paragraph on the effects of warming on plant reproduction more concise, as suggested.

Section 5.1 – while I agree with the overall thrust of what is said here, it has to be recognised that these are mostly general statements of what ‘could’ happen, and each represents quite a large logical ‘jump’ from the actual data obtained and presented here.

Current L392-410. We have made fewer broad leaps in this paragraph, and we have removed the speculative final sentence.

L440 – you can only say this, as noted above, if these species definitely were not present in the study plots at the start of the experiment. And even if not, given the very small number of instances, I can’t see that you can unequivocally tie the occurrence now to being caused by the experimental warming.

Current L413-32. We have removed the discussion of these species, as well as the discussion of bryophyte cover differences between treatments (which we have removed from the manuscript, see above).

L448 – same comment, did it decrease relative to the original state in these plots?

Current L413-32. We have removed the discussion of bryophyte cover differences between treatments (which we have removed from the manuscript, see above).

L477 – as noted above, the subject in this paragraph (which is also more extended than it could be) is not really justified by the strength of the data presented in this study.

Current L413-432. We made this paragraph shorter and much less speculative. We have now combined it with the previous paragraph. We have explained the limitation of our findings and also placed them more in context.

L498 – the paper that forms part of Day’s overall warming study reporting detailed analyses of responses of contained invertebrate communities is Convey et al. (2002, Ecology), which is referred to later in the same paragraph here.

*Current L439. The Day et al. 2009 paper is titled “Response of plants and the dominant microarthropod, *Cryptopygus antarcticus*, to warming and contrasting precipitation regimes in Antarctic tundra” and discusses the invertebrate community, but focuses on one species. We have clarified this in the Discussion.*

Nb that reference formatting in these paragraphs at this point in the ms differs from that elsewhere.

We have fixed the formatting of the references in the text.

L512 – note that the Mouratov et al paper contains entirely incorrect identifications

of the nematodes present on King George Island, erroneously based on a key from continental Antarctic Victoria Land. However, there are some very thorough studies of soil nematodes from Victoria Land by US researchers that document different soil water and salt ion preferences/tolerances of the nematode species that occur there.

We have clarified this point.

Para at l517 – this another example of a long paragraph that in reality is largely speculative. In the absence of invertebrate identifications to species (or genus in nematodes; and note above comment about the fundamental error in identification and hence interpretation in Mouratov et al) it is virtually impossible to justify the sort of statements made in this paragraph, other than as unsubstantiated generalisations.

Current L457-473. We have made this paragraph less speculative, including our speculation about the nematodes.

L558 – this is another overstatement of this point! The study very clearly did not observe any such change in reproductive output, as no sporophytes were observed at all!

Current L489. We agree that this needs to be changed. We have replaced “reproductive output” with “sexual expression.”

References – note miss-spelling of Bokhorst at points (not Bokhurst). There are clear formatting inconsistencies and errors throughout the ref list.

We have edited the references.

Reviewer: 2

Comments to the Author(s)

This paper draws attention by suggesting data that may demonstrate the behavior of Antarctic plants in the face of environmental changes, especially warming in Antarctica.

However, the information presented is still very superficial, only two species were effectively evaluated. Therefore, se tem OTCs in several others sites in King George Island, why don't include the data of these? Why other patches of different species were no evaluate also? Based all the discussion on these couple species could be configure an over extrapolation of the results.

We have made our conclusions less speculative, as suggested by both reviewers. These two plant species are the dominant species in these OTC, and with restricted sampling to limit damage to the plots, these species are what we were able to sample. Our other

OTCs are part of a large international collaboration, and we were only able to collect samples from this site. However, the collaboration does allow us to look at OTC that have been established for eight years.

The reproductive biology methods does not sound right to do. The age of moss caulonemata its too difficult to define only by colours. And as well, as an extremely dry environment, seek for sporophytes is quite be an challenge, mainly the female gamethophores will be found frequently. Its is an major issue that could unworthy the paper results.

Current L205-207. Using color and morphology is the common protocol to place bryophyte leaves/shoots in age classes, and we have now provided additional information and the appropriate references.

Current L207-215. Our methodology for identifying sporophytes is robust, as partial seta remain after spores are released. This methodology is widely used in bryophytes, including in dry environments. We have previously assessed both species for sporophytes in Antarctica (Casanova-Katny et al. 2016 Revista Chilena de Historia Natural) and have examined herbarium specimens at the British Antarctic Survey Herbarium (Cambridge, UK) to assess the range of sporophytes on samples in this region of Antarctica. We are confident in this methodology and our result that no sporophytes were present in our cores.

Current L207-215. The methods we used for quantifying male and female sex organs are standard for bryophytes, including for species in this plant family, and are extremely robust. We dissect shoots under the microscope, so there is no bias in finding sex organs. We have now included additional information and references about these methods. We have used these methods to quantify males and female sex organs in Antarctica (Shortlidge et al. 2017 Annals of Botany), and in other environments (e.g., Eppley et al. 2011 International Journal of Plant Sciences; Slate et al. 2017 Annals of Botany). We are confident in our methodology and results.

The section with the most security in the presentation of results and discussions is the one that deals with comparing the anthropofauna associated with the moss spots. There I believe they have a more robust and exploitable result. I suggest to re-write the paper concerning only in such founds.

We have made the Discussion more concise (removing discussion on the effect of warming on individual plant species percent cover) and less speculative, as suggested by Reviewer 1, and feel that this also focuses the results more clearly on the invertebrate results, as suggested by Reviewer 2

Appendix B

5 October 2019

Dear Professor Kevin Padian
Open Science, Subject Editor

Dear Dr. Padian:

We are resubmitting our manuscript "Species-specific effects of passive warming in an Antarctic moss system" (RSOS ID 190744) for publication in *Open Science*. This manuscript was accepted with minor revisions. In this revised manuscript, you will find that we have carefully addressed the reviewers' comments. In particular, we added a supplemental photo figure, as suggested. We have made the other minor changes suggested and have indicated our point-by-point responses below (in italics).

Thank you for your patience, time and consideration, and we appreciate the careful consideration of our manuscript by the reviewers, which has greatly improved it. We look forward to hearing from you at your earliest convenience.

Sincerely,
Hannah Prather

Associate Editor Comments to Author:

The reviewers and editors are grateful for the efforts you've made to improve the manuscript; however, they identify a number of changes necessary to get the paper over the line. Please carefully respond to the reviewers' recommendations before you resubmit.

Reviewer comments to Author:

Reviewer: 2

Comments to the Author(s)

Re-reading the manuscript sent by the authors, it is noted that an extensive improvement in the arguments was made, giving greater reliability to the results. However, I believe that it is necessary to give due clarity to the experiment. The authors argue that they used the usual methodology for specimen identification and reproductive organ evaluation in *P. alpinum* samples. In other words, they had access to physical material for gametophyte phenotyping analysis. What I would like to see, and would give more transparency to the data, would be images of these gametophytes, in their habit, to illustrate to readers that they are even these two species (mainly for *S. georgicouncinata* which is strongly disputed from the point of view). species circumscription), as well as images of the archegons identified in the sexual expression analyzes. On the other hand, would you like to know if male individuals were found outside OTCs? Those who explore and research the Antarctic vegetation, especially those that go on for more than 20 years annually, report no sporophytes of some species, and polytrichaceas are among them (if I'm not mistaken the last finding was in the 1990s, at work). published by Peter Convey and Prof. R. Lewis-Smith). Much is still theorized about the advantage of vegetative rather than sexual reproduction for these species because of the extreme climate. But it can also be thought of because there's not so many individuals representing both sexes in the same compartment (In Admiralty Bay, for example, it is recurrent to find male individuals with splash-cups, and few or almost no female individuals), or about decreased water availability in areas where vegetation grows in Antarctica, so I would like to see a sign of this at work, if only to show evidence that male individuals were found outside the OTC, including images if they are still available.

*As suggested, we have provided a figure (Supplementary Figure 1) showing the plants on King George Island, and also showing female archegonia from *P. alpinum* from a control plot. We did not observe sporophytes or males during this experiment, and this is noted in the manuscript (current L278-279; 341-342). Gametophytes were dissected in the laboratory to determine presence of sexual structures (current L208-216).*

Regarding the modifications requested by the reviewer I, I believe that this one will also manifest, but I think it still needs to tie up the different associated experiments. I can understand all the reasons that led the authors to perform these experiments, but I had the happy opportunity to read and follow two different moments in the

development of this work, and the reader who will access the final article after the acceptance will not have this same opportunity. So I believe there is still some information (unfortunate in the introduction) that can tie the issue of heating, with the sex ratio and the arthropods.

Based on the previous feedback of Reviewer 1 and Reviewer 2, we feel that we have achieved a balance to meet the requested reductions and clarifications to the Results and Discussion sections, and we feel the manuscript revisions were overall fairly modest, providing a still cohesive manuscript.

Reviewer: 1

Comments to the Author(s)

Species-specific effects of passive warming in an Antarctic moss system

Hannah M. Prather et al (revision)

The authors have made good efforts to respond to the matters raised by both reviewers, and I am satisfied with the responses and overall changes. Reading the new version of the ms raises the following minor issues:

L19 – delete ‘regions’ – the sentence doesn’t read correctly as it is currently

Current L19. This has been corrected.

L26 –‘dominant’ not ‘dominate’

Current L26. This has been corrected.

L29-31 – the concept of ‘reproductive shifts’ and particularly ‘reproduction’ is still stated too strongly here – there was no change in actual reproduction in the expt (ie no sporophytes), and this idea needs to be de-emphasised.

Current L28-29. We have de-emphasized the concept of ‘reproductive shifts’ to clarify that our results show evidence of shifts in sexual expression.

L34 – what does ‘differentially impact’ mean in the context of this sentence? I think I would reduce this to ‘may impact...’

Current L34. We have incorporated this suggestion.

L55 – ‘Signy’ is miss-spelt

Current L55. This has been corrected.

L100 – ‘Antarctic’

Current L100. This has been corrected.

L101-4 – the statement about the current lack of these studies is correct but, again, in that the current study did not find any form of change in actual reproduction or in reproductive output, in essence this is a subject that the current study cannot actually address – I do not think it is appropriate to say, in effect, ‘if we had found any changes then this study would have addressed this subject’!

Current L103. We have modified this sentence to reflect that we were examining reproductive expression and not overall reproductive output in these

L106 – what is ‘Antarctic terrestrialization’?

Current L106-107. We have re-worded this sentence to add clarification.

L121 – archipelago should not be capitalized

Current L121. This has been corrected.

L123 – delete ‘unique’ – any confirmed species a unique species in some sense.

Current L123. This suggestion has been incorporated.

L132 – ‘sexually reproductive’ – plainly they frequently reproduce asexually

Current L132. This suggestion has been incorporated.

L170 – ‘recordings were made’

Current L170. This has been corrected.

L280 – oribatid should not be capitalised, also at l346 and l358 (the proper name of the group, Oribatida, would be)

Current L281, L355, L359, L471. These corrections have been made.

L283 – ‘and the number’

Current L283. This has been corrected.

L298-300 – it seems odd that all three ANOVAs have the same F value here, especially as there are two different p values as well but identical sample sizes (and again at l303/4)?

Current L299-304. We have corrected one error and clarified the reporting of the statistics here to reduce the redundancy.

L386 – ‘higher trophic levels’ – some caution needed here and elsewhere in the ms, as some readers will take this as implying that the mites, springtails or nematodes actively graze on the mosses (producers), which they do not. Rather they are microbivores – if they graze on living algae (likely for the springtails) then they can be considered primary consumers. But many will rather be detritivores, grazing decaying dead plant material and the bacterial/fungal components therein. There are certainly not plural higher trophic levels examined in this study.

Current L387-388. We agree with this comment and have made the suggested edit to clarify the point.

L394 – control

Current L395. This has been corrected.

L439 – picking up on one of the review responses, the title of the Day et al 2009 is indeed as referred, but the primary research paper reporting the Collembola study which it summarises, and which took place as part of the Day group’s overall project, is that of Convey et al. (2002).

Current L442-445. The findings of Convey et al. (2002) are discussed here.

L441 – capitalise ‘Peninsula’

Current L442. This has been corrected.

L444 – delete ‘endemic’ and capitalise ‘Collembola’

Current L445. This has been corrected.

L470 – capitalise ‘Oribatida’ and ‘Collembola’; also the general statement that these micro-arthropods are fungivores is an overstatement of the information the cited reference gives. In fact, many springtails and in particular the commonly dominant *Cryptopygus antarcticus* are algivores rather than fungivores (see also earlier comment on whether these species can be more appropriately be considered in the decomposition cycle rather than as ‘higher trophic levels’).

Current L470. We have clarified this point and updated the citation utilized here.

L514 – Chilean Antarctic Institute

Current L515. This has been corrected.